# MIND'S EYE: IMAGE RECOGNITION BY EEG VIA MULTIMODAL SIMILARITY-KEEPING CONTRASTIVE LEARNING

## ABSTRACT

Decoding images from non-invasive electroencephalographic (EEG) signals has been a grand challenge in understanding how the human brain process visual information in real-world scenarios. To cope with the issues of signal-to-noise ratio and nonstationarity, this paper introduces a MUltimodal Similarity-keeping contrastivE learning (MUSE) framework for zero-shot EEG-based image classification. We develop a series of multivariate time-series encoders tailored for EEG signals and assess the efficacy of regularized contrastive EEG-Image pretraining using an extensive visual EEG dataset. Our method achieves state-of-the-art performance, with a top-1 accuracy of 19.3% and a top-5 accuracy of 48.8% in 200-way zero-shot image classification. Furthermore, we visualize neural patterns via model interpretation, shedding light on the visual processing dynamics in the human brain.

## 1 INTRODUCTION

Understanding visual processing in the human brain remains a profound challenge at the intersection of neuroscience and artificial intelligence. Visual processing involves a complex sequence of neural mechanisms across various brain regions, enabling the intricate processing of visual stimuli Riesenhuber & Poggio (1999); Miyawaki et al. (2008); Liu et al. (2009); DiCarlo et al. (2012); Gifford et al. (2022). The development of deep learning techniques, such as convolutional neural networks (CNNs), has been significantly inspired by our understanding of these neural mechanisms Fukushima (1980); LeCun et al. (1998; 2015). Unveiling the brain dynamics of visual processing in real-world contexts holds the potential to inspire future advancements in artificial intelligence (AI), continuing the cycle of innovation driven by biological insights Hassabis et al. (2017); Ullman (2019). Recent studies have advanced our understanding of visual processing in the human brain through the observation of brain activity using various neuromonitoring modalities He et al. (2011). Electroencephalography (EEG), as a non-invasive, portable modality with high-temporal resolution, offers a unique window into visual processing by revealing the instantaneous neural dynamics of visual perception and recognition in real-world contexts Rousselet et al. (2007); Samaha & Postle (2015); Wei & Jung (2023).

Decoding images from EEG signals represents a promising approach to study the mechanisms of visual processing. By leveraging EEG, researchers can gain insight into the temporal evolution of neural responses to visual stimuli Robinson et al. (2017). However, this endeavor faces significant obstacles, primarily due to the low signal-to-noise ratio and nonstationarity of EEG signals Kaplan et al. (2005); Urigüen & Garcia-Zapirain (2015). Addressing these challenges is crucial for advancing our understanding of visual cognition and for developing robust EEG-based image decoding or brain-computer interfacing (BCI) systems. Early studies in EEG-based image decoding have been constrained by the use of small datasets, limiting their ability to develop generalizable models Spampinato et al. (2017); Tirupattur et al. (2018). More recent work has utilized larger datasets collected through the rapid serial visual presentation (RSVP) paradigm, where images are presented in quick succession to elicit brain responses Gifford et al. (2022); Song et al. (2024). Despite these advances, the performance of existing methods remains suboptimal, underscoring the need for dedicated design of EEG encoding network architectures that consider the brain's mechanisms and EEG characteristics.

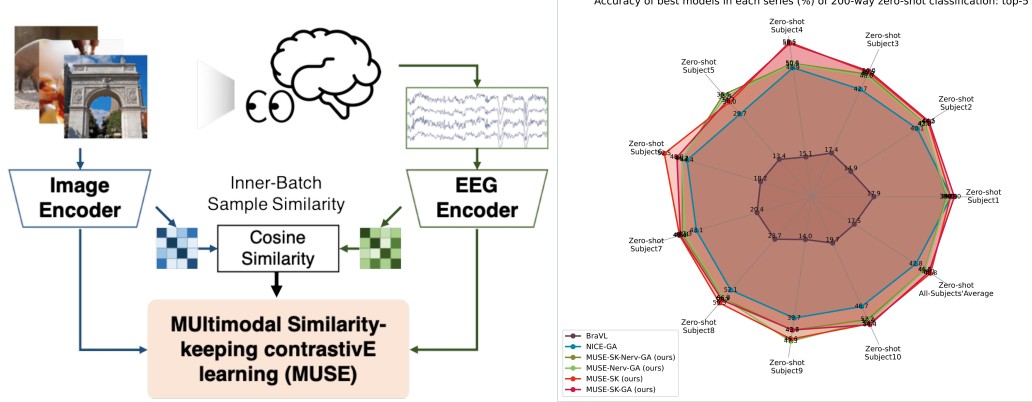

Figure 1: Schematic illustration of the proposed MUltimodal Similarity-keeping contrastivE learning (MUSE) framework. During the training phase, EEG-image pairs are independently processed by an EEG encoder and an image encoder. The objectives of the MUSE framework are twofold: 1) maximize the separation between matched and unmatched pairs, and 2) maintain the inner-batch sample similarity within each EEG-image pair (see Algorithm 1 for details). In the test phase, an unseen EEG sample is passed through the EEG encoder, which identifies the most similar image from a set of unseen images based on cross-modality embedding similarity.

To address the challenges in EEG-based image decoding, we present a novel self-supervised framework, coined as multimodal similarity-keeping contrastive learning (MUSE), dedicated to cross-modality contrastive learning between EEG and image data. We develop a series of multivariate time-series encoder network architectures tailored for EEG processing that facilitate the cross-modality contrastive learning with an advanced off-the-shelf image encoder (CLIP-ViT Radford et al. (2021)). These encoders feature an upstream spatial convolution of EEG data for the sake of feature extraction and noise suppression Wei et al. (2019); Pan et al. (2022). Additionally, we propose an innovative similarity-keeping contrastive learning mechanism, inspired by the cortical mapping organization of visual object representation in the inferotemporal (IT) cortex Bao et al. (2020), to regularize the contrastive learning process using the information of inter-object relationships within both EEG and image samples.

Furthermore, we employ model interpretation techniques to visualize the neural patterns of image processing, offering a deeper understanding of the underlying dynamics of visual cognition in the human brain. The contributions of this work are threefold:

- We introduce a novel self-supervised multimodal similarity-keeping contrastive learning (MUSE) framework that achieves state-of-the-art performance in zero-shot EEG-based image recognition.
- We propose EEG encoders with upstream spatial convolution and similarity-keeping regularization to enhance EEG-image cross-modality contrastive learning.
- We visualize neural patterns through model interpretation to provide neuroscientific insights into the spatial and temporal brain dynamics of visual processing.

## 2 RELATED WORKS

### 2.1 DECODING VISUAL INFORMATION FROM BRAIN SIGNALS

Interpreting visual data from the human brain has been a longstanding challenge at the intersection of neuroscience and computer science Riesenhuber & Poggio (1999); Miyawaki et al. (2008); DiCarlo et al. (2012); Gifford et al. (2022). Despite significant advancements in understanding static visual inputs, rapidly and accurately extracting meaningful information from natural imagery remains difficult Kay et al. (2008); Chen et al. (2023). Previous efforts have primarily utilized functional magnetic resonance imaging (fMRI) Mai et al. (2023); Takagi & Nishimoto (2023); Scotti et al.

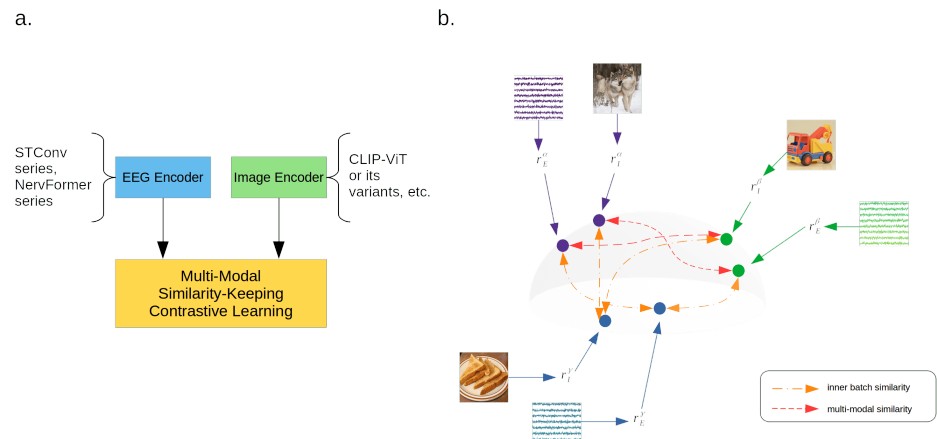

Figure 2: (a.) The whole view of this work. (b.) Illustration on feature space of multimodal similarity-keeping contrastive learning framework (MUSE), different from traditional contrastive learning only focus on multimodal similarity, MUSE both consider the multimodal similarity and inner batch similarity in the loss function. $r$ denotes representation. $I$ and $E$ denotes image and EEG signal, respectively.

(2024), which has demonstrated the ability to capture meaningful content and structural details from visual processing in the brain. However, fMRI relies on detecting changes in blood oxygenation, resulting in a temporal lag of several seconds per stimulus, thereby limiting its utility for real-time applications. Additionally, fMRI is expensive and requires large, stationary equipment.

In contrast, electroencephalography (EEG) offers superior temporal resolution, immediate data feedback, and portable, cost-effective hardware. These attributes position EEG as a promising candidate for personal brain-computer interface technology. Nevertheless, current methods for using EEG to extract semantic information for image classification have not achieved satisfactory results Ahmed et al. (2021); Liu et al. (2023); Song et al. (2024), highlighting the need for improved approaches. Previous methodologies have often relied on supervised learning techniques with a limited set of image categories, ignoring the intrinsic correlations between visual stimuli and neural responses Liu et al. (2023); Spampinato et al. (2017); Singh et al. (2024). These limitations impair their effectiveness in real-world scenarios that require the generalization to recognize novel, unfamiliar object categories. To address these issues, Du et al. (2023) first attempted zero-shot classification using the largest available EEG-image database Gifford et al. (2022) with a multilayer MLP and joint EEG-image-text representation, while Song et al. (2024) employed a contrastive learning method. However, Song et al. (2024) utilized a basic contrastive learning framework based on CLIP Radford et al. (2021). Our work improves upon this framework and the EEG encoder, introducing a self-supervised learning approach for EEG-based image decoding. This framework allows the model to generalize to object recognition tasks without specific prior training, demonstrating its effectiveness.

## 2.2 MULTIMODAL CONTRASTIVE LEARNING

In recent years, after the success of the traditional contrastive learning models on the same modal data like text and image Tian et al. (2020); He et al. (2020); Grill et al. (2020); Chen et al. (2020), the development of multimodal contrastive learning has reached significant advancements in the field of self-supervised learning, particularly in tasks that contain the integration of multiple types of data. This method leverages the strengths of various modalities (e.g., text, images, video) to boost model generalization across diverse datasets. Multimodal contrastive learning aligns representations from different modalities within a shared embedding space, facilitating robust, modality-invariant feature learning. This enhances capabilities in cross-modal retrieval and zero-shot learning. Typically, a two-tower network architecture processes each modality independently, with outputs converging in the embedding space where contrastive loss minimizes distances between similar pairs and maximizes distances between dissimilar ones. One of the most popular and successful multimodal contrastive learning framework is CLIP Radford et al. (2021), which project both the image and text in to the

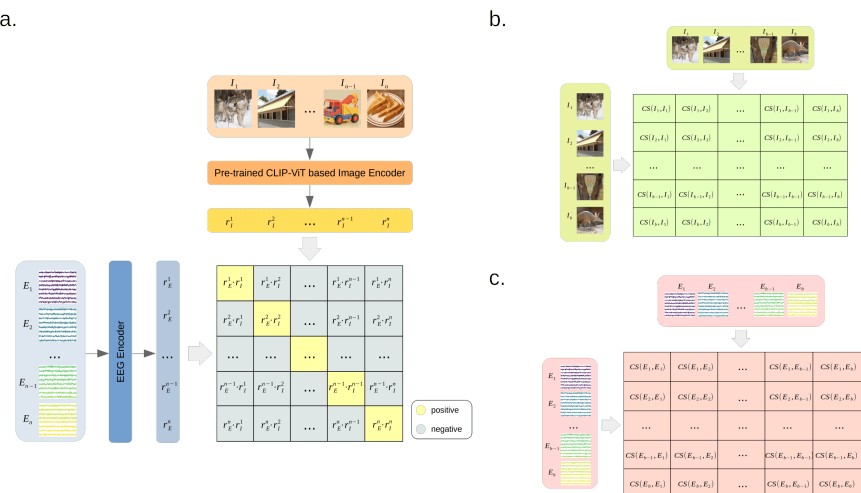

Figure 3: The details of the MUSE. (a.) The contrastive learning loss is calculated from EEG encoding and image encoding. (b.)(c.) The similarity-keeping loss comes from the final similarity of self-batch similarity of the input modal data.

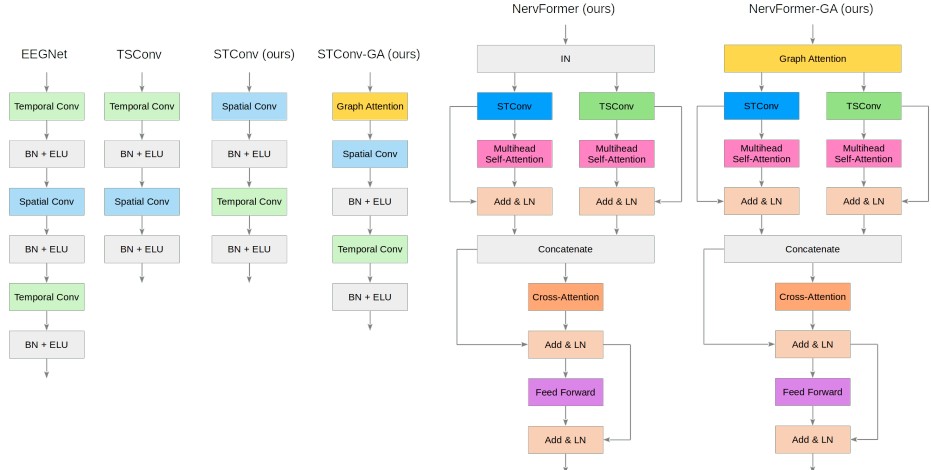

Figure 4: The model structure comparison. Where BN denotes batch normalization, IN denotes instance normalization, LN denotes layer normalization, respectively.

same feature space. Nevertheless, because datasets containing both time-series signals like EEG and image data are quite rare, there has been little research applying contrastive learning methods to this combination of temporal and visual information. To our best knowledge, Ye et al. (2022) is maybe the first work introduced the EEG-image contrastive learning on obtaining the EEG-image representation for image reconstruction downstream task but do not do the zero-shot classification. Singh et al. (2024) introduced the EEGClip network for joint representation learning between EEG signal and image but it just do supervised learning. Song et al. (2024) first try to design the EEG encoder on EEG-image contrastive learning , but the work only modified the encoders. This area remains largely uncharted and calls for new, specialized contrastive learning techniques to handle these joint time-series and image modalities effectively.

---

**Algorithm 1** Multimodal Similarity-Keeping Contrastive Learning framework (MUSE)

---

1: **Input**: (Image, EEG)          ▷ stimulus & response
2: **Model**: $Enc_{img}$: CLIP-ViT or its variance, $Enc_{eeg}$: STConv or NervFormer

3: # E : $(batch, channel, electrode, data\ sample)$      ▷ batch of input EEGs
4: # I : $(batch, channel, height, width)$      ▷ batch of input images

5: # $\tau$ : learned temperature parameter
6: # $\beta$ : learned inner similarity parameter
7: # CS : Cosine Similarity
8: # SK : Similarity-Keeping

9: # extract normalized representations from the raw image and EEG
10: $E_f = \text{Norm}(\text{Linear}(Enc_{eeg}(\text{E})))$
11: $I_f = \text{Norm}(Enc_{img}(\text{I}))$      ▷ can be obtained before training

12: # calculate cosine similarity from the inner batch image and EEG
13: $E_{CS} = \text{CS}(E_f, E_f)$
14: $I_{CS} = \text{CS}(I_f, I_f)$
15: $loss_{SK} = 1 - \mathbb{E}(\text{CS}(E_{CS}, I_{CS}))$

16: # scaled pairwise cosine similarity
17: logits = $\text{dot}(E_f, I_f.\text{t}) \times e^{\tau}$

18: # symmetric loss function
19: labels = arange(batch)      ▷ self-supervised learning label
20: $loss_e = \text{CrossEntropyLoss(logits, labels, axis=0)}$
21: $loss_i = \text{CrossEntropyLoss(logits, labels, axis=1)}$
22: $total\_loss = (loss_e + loss_i)\ /\ 2 + \beta \times loss_{SK}$

---

# 3 METHODOLOGY

## 3.1 OVERVIEW

This section introduces the Multimodal Similarity-Keeping Contrastive Learning (MUSE) framework, comprising the EEG encoder, image encoder, and the contrastive learning method. Our contribution encompasses cutting-edge EEG encoders tailored for zero-shot classification tasks: the Spatial-Temporal convolution (STConv) and NervFormer architectures, along with a pioneering regularized contrastive learning approach featuring a novel similarity-keeping loss.

## 3.2 NETWORK ARCHITECTURE

### 3.2.1 EEG ENCODER

In this study, we introduce a series of multivariate time-series encoding architectures tailored to capture essential features in EEG data. Recent works suggest that upstream spatial convolution serves as an effective spatial filtering method for enhancing feature extraction and noise suppression Wei et al. (2019); Pan et al. (2022). Herein, we present the Spatial-Temporal Convolution (STConv) module, which employs spatial convolution to denoise data by referencing between brain electrodes, followed by temporal convolution. Additionally, we extend the capabilities of the STConv and Temporal-Spatial Convolution (TSConv) modules by integrating an attention mechanism, leading to the development of a novel transformer-like EEG encoder, which we refer to as NervFormer. In line with Graph Attention Networks (GATs) principles, we employ the Graph Attention (GA) module (see Appendix) to iteratively refine the state of each node, conceptualized as electrodes, by leveraging the states of all other nodes Veličković et al. (2018); Brody et al. (2022). The architectures of the baseline and proposed EEG encoders are illustrated and compared in Figure 4.

### 3.2.2 IMAGE ENCODER

For our implementation, we integrate the off-the-shelf CLIP-ViT model Radford et al. (2021), which has demonstrated exceptional performance in aligning image and text representations. This model, pre-trained on extensive datasets, captures intricate details and high-level semantic information from images, making it an ideal candidate for our contrastive learning framework.

### 3.2.3 SIMILARITY-KEEPING CONTRASTIVE LEARNING

Inspired by recent neuroscience findings of the cortical network of visual object representation Bao et al. (2020); She et al. (2024), we take the interplay between object categories into account and propose a novel regularized contrastive learning framework. The procedure is outlined in Algorithm 1.

The ordinary contrastive learning uses InfoNCE loss given by Oord et al. (2018); He et al. (2020); Radford et al. (2021):

$$\mathcal{L}_{InfoNCE} = -\mathbb{E}\left[\log \frac{\exp(S_{E,I}/\tau)}{\sum_{k=1}^{N} \exp(S_{E,I_k}/\tau)}\right] \tag{1}$$

where the $S_{E,I}$ denotes the similarity score between EEG signal $E$ and image $I$ pairing data, the $\tau$ is learned temperature parameter, the training process shown in Figure 2.

We introduce regularization to the ordinary contrastive learning by incorporating similarity preservation into the contrastive loss to capture both inter-sample and multimodal similarities. Drawing inspiration from the similarity-keeping (SK) concept used in knowledge distillation between EEG models Huang et al. (2023), we propose a novel SK loss to regularize the InfoNCE loss. This involves estimating the inner-batch inter-sample relationship. The SK loss is defined as:

$$\mathcal{L}_{SK} = 1 - \mathbb{E}\left[S(S_{E,E}, S_{I,I})\right] \tag{2}$$

We introduce a trainable parameter $\beta$ to enhance training flexibility. When the $\beta = 0$, the similarity-keeping InfoNCE loss reduces to the standard InfoNCE loss. The combined loss function, which integrates similarity-keeping, is illustrated in Figure 3 and defined as:

$$\mathcal{L}_{SK-InfoNCE} = \mathcal{L}_{InfoNCE} + \beta \times \mathcal{L}_{SK} \tag{3}$$

This integration of similarity-keeping into the contrastive loss framework ensures that the model not only aligns paired EEG and image embeddings effectively but also maintains the intrinsic relationships within the batch.

## 4 EXPERIMENTS

### 4.1 DATASETS AND PREPROCESSING

The ThingsEEG dataset Gifford et al. (2022) comprises extensive EEG recordings gathered through a rapid serial visual presentation (RSVP) paradigm, featuring responses from 10 individuals to 16,740 natural images from the THINGS database Hebart et al. (2019). The dataset includes 1654 training classes, each with 10 images, and 200 test classes, each with 1 image. EEG recordings were conducted using 64-channel EASYCAP equipment, and the data were preprocessed by segmenting into trials from 0 to 1000 ms post-stimulus onset, with baseline correction using the pre-stimulus mean. EEG responses for each image were averaged across repetitions, and the images were resized to 224×224 and normalized prior to processing.

### 4.2 EXPERIMENT DETAILS

Experiments were conducted on a GeForce RTX 3090 24G GPU with Pytorch. Training using the MUSE series required approximately 2 to 3 hours per subject, with a batch size of 1000, while

Table 1: Overall accuracy (%) of 200-way zero-shot classification using CLIP-ViT as image encoder: top-1 and top-5. The parts in bold represent the best results, while the underlined parts are the second best.

| Method | Subject 1 | | Subject 2 | | Subject 3 | | Subject 4 | | Subject 5 | | Subject 6 | | Subject 7 | | Subject 8 | | Subject 9 | | Subject 10 | | Ave | |
|---|---|---|---|---|---|---|---|---|---|---|---|---|---|---|---|---|---|---|---|---|---|---|
| | top-1 | top-5 | top-1 | top-5 | top-1 | top-5 | top-1 | top-5 | top-1 | top-5 | top-1 | top-5 | top-1 | top-5 | top-1 | top-5 | top-1 | top-5 | top-1 | top-5 | top-1 | top-5 |
| Subject dependent - train and test on one subject | | | | | | | | | | | | | | | | | | | | | | |
| BraVL | 6.1 | 17.9 | 4.9 | 14.9 | 5.6 | 17.4 | 5.0 | 15.1 | 4.0 | 13.4 | 6.0 | 18.2 | 6.5 | 20.4 | 8.8 | 23.7 | 4.3 | 14.0 | 7.0 | 19.7 | 5.8 | 17.5 |
| NICE | 12.3 | 36.6 | 10.4 | 33.9 | 13.1 | 39.0 | 16.4 | 47.0 | 8.0 | 26.9 | 14.1 | 40.6 | 15.2 | 42.1 | 20.0 | 49.9 | 13.3 | 37.1 | 14.9 | 41.9 | 13.8 | 39.5 |
| NICE-SA | 13.3 | 40.2 | 12.1 | 36.1 | 15.3 | 39.6 | 15.9 | 49.0 | 9.8 | 34.4 | 14.2 | 42.4 | 17.9 | 43.6 | 18.2 | 50.2 | 14.4 | 38.7 | 16.0 | 42.8 | 14.7 | 41.7 |
| NICE-GA | 15.2 | 40.1 | 13.9 | 40.1 | 14.7 | 42.7 | 17.6 | 48.9 | 9.0 | 29.7 | 16.4 | 44.4 | 14.9 | 43.1 | 20.3 | 52.1 | 14.1 | 39.7 | 19.6 | 46.7 | 15.6 | 42.8 |
| MUSE-Nerv (ours) | 11.0 | 33.9 | 12.3 | 37.4 | 13.6 | 39.4 | 19.1 | 48.0 | 10.7 | 31.9 | 14.0 | 41.2 | 13.0 | 41.3 | 21.0 | 54.6 | 15.4 | 38.6 | 17.1 | 43.9 | 14.7 | 41.0 |
| MUSE-SK-Nerv (ours) | 11.6 | 34.7 | 14.3 | 40.4 | 13.6 | 38.2 | 20.8 | 48.6 | 12.0 | 32.2 | 16.1 | 41.5 | 15.7 | 43.7 | 24.1 | 54.4 | 17.2 | 41.7 | 17.1 | 44.7 | 16.3 | 42.0 |
| MUSE-SK-Nerv-GA (ours) | 12.1 | 38.7 | 15.2 | 43.0 | 18.5 | 48.8 | 24.4 | 50.6 | 14.0 | 36.6 | 18.0 | 46.1 | 19.7 | 48.4 | 24.3 | 56.9 | 17.8 | 43.7 | 21.9 | 52.2 | 18.6 | 46.5 |
| MUSE-Nerv-GA (ours) | 13.4 | 39.0 | 17.6 | 42.8 | 17.3 | 48.0 | 22.6 | 50.3 | 14.4 | 35.9 | 18.7 | 46.2 | 19.2 | 47.3 | 26.8 | 56.7 | 19.0 | 47.3 | 20.6 | 52.9 | 19.0 | 46.6 |
| MUSE (ours) | 14.7 | 39.2 | 15.2 | 45.3 | 19.3 | 48.7 | 25.9 | 61.0 | 12.6 | 36.0 | 18.5 | 50.6 | 20.2 | 50.1 | 26.3 | 58.6 | 19.0 | 45.7 | 20.4 | 54.0 | 19.2 | 48.9 |
| MUSE-GA (ours) | 14.7 | 38.3 | 17.5 | 47.4 | 17.1 | 48.0 | 24.8 | 58.2 | 11.5 | 34.9 | 18.5 | 50.5 | 19.3 | 49.1 | 24.3 | 55.1 | 16.9 | 40.3 | 24.0 | 55.8 | 18.8 | 47.8 |
| MUSE-SK (ours) | 14.4 | 39.9 | 16.5 | 44.2 | 19.7 | 49.5 | 26.4 | 58.6 | 13.2 | 34.0 | 19.1 | 52.5 | 19.5 | 49.4 | 26.8 | 59.3 | 17.6 | 46.6 | 20.1 | 54.3 | 19.3 | 48.8 |
| MUSE-SK-GA (ours) | 15.3 | 41.0 | 18.1 | 44.5 | 20.0 | 50.0 | 25.3 | 58.1 | 11.2 | 34.7 | 17.9 | 48.0 | 20.1 | 49.1 | 25.4 | 57.7 | 17.0 | 43.6 | 22.7 | 54.4 | 19.3 | 48.1 |

NervFormer series models took 40 minutes to 1 hour per subject. Models were saved at 200 epochs when the validation loss reached its lowest point. We use the weighted Adam optimizer with a learning rate of 0.0002 and parameters $\beta_1$=0.5 and $\beta_2$=0.999. The $\tau$ in contrastive learning initialized with $log(1/0.07)$ and $\beta$=1. The NervFormer model achieves the best results with a multiheads number of 5. Results were averaged over five random seeds, and statistical significance was determined using the Wilcoxon Signed-Rank Test.

## 4.3 PERFORMANCE COMPARISON

The comparison results presented in Table 1 highlight the performance of various methods, with detailed model abbreviations provided in the appendix. Overall, MUSE-SK achieves the highest average top-1 accuracy at 19.3%, while MUSE attains the highest average top-5 accuracy at 48.9%. Furthermore, MUSE-SK-Nerv-GA, MUSE-Nerv-GA, MUSE, MUSE-SK, MUSE-SK-GA, MUSE-GA, and MUSE-SK-Nerv-GA significantly outperform the NICE-GA model in both top-1 ($p < 0.01$) and top-5 ($p < 0.01$) accuracy. Although individual performance can differ, MUSE-based methods usually do better than others. The GA and SK variants are particularly strong in this evaluation.

## 4.4 ABLATION STUDY

We conduct ablation studies on both MUSE and MUSE-Nerv series models, with the results of MUSE-Nerv illustrated in Table 3. While the NervFormer EEG encoder does not demonstrate the best average zero-shot performance across all datasets, the MUSE-SK-Nerv-GA model achieves higher individual accuracy for subjects 5 and 10 compared to both MUSE and MUSE-SK. Moreover, beyond the MUSE series models, which solely employ the STConv as the EEG encoder, the MUSE-Nerv series models, incorporating the NervFormer as the EEG encoder, independently validate the efficacy of the similarity-keeping loss architecture and the graph attention module in EEG-image multimodal contrastive learning.

Upon examining the performance metrics of MUSE as depicted in Table 2, it becomes apparent that MUSE, MUSE-SK, and MUSE-SK-GA exhibit similar average performance levels. However, each method demonstrates distinct advantages across the ten subjects studied. For example, MUSE-SK-GA demonstrates superior overall performance in subjects 1, 3, and 10, while MUSE-SK achieves state-of-the-art results in subject 8. Additionally, each method excels uniquely in either top-1 or top-5 rankings in various subjects. This underscores the effectiveness of the SK and GA techniques as enhancements. However, in the context of STConv, these techniques do not demonstrate as clear an advantage as NervFormer does. We also observe that while SK may impact GA performance on NervFormer, both SK and GA enhance performance on STConv, with further details discussed in the model interpretation section.

## 4.5 MODEL INTERPRETATION

We conducted model interpretation to uncover the internal mechanisms of our models across three distinct domains: spatial-temporal, brain region topography-temporal, and temporal-frequency. We

Table 2: Ablation Study of MUSE series models, accuracy (%) of 200-way zero-shot classification: top-1 and top-5. The parts in bold represent the best results, while the underlined parts are the second best.

| Method | Subject 1 | | Subject 2 | | Subject 3 | | Subject 4 | | Subject 5 | | Subject 6 | | Subject 7 | | Subject 8 | | Subject 9 | | Subject 10 | | Ave | | Win |
| --- | --- | --- | --- | --- | --- | --- | --- | --- | --- | --- | --- | --- | --- | --- | --- | --- | --- | --- | --- | --- | --- | --- | --- |
| | top-1 | top-5 | top-1 | top-5 | top-1 | top-5 | top-1 | top-5 | top-1 | top-5 | top-1 | top-5 | top-1 | top-5 | top-1 | top-5 | top-1 | top-5 | top-1 | top-5 | top-1 | top-5 | subject score # |
| Subject dependent - train and test on one subject | | | | | | | | | | | | | | | | | | | | | | | |
| *Original MUSE (STConv as EEG encoder & CLIP-ViT as image encoder with InfoNCE loss)* | | | | | | | | | | | | | | | | | | | | | | | |
| MUSE | 14.7 | 39.2 | 15.2 | **45.3** | 19.3 | 48.7 | 25.9 | **61.0** | 12.6 | **36.0** | 18.5 | 50.6 | 20.2 | 50.1 | 26.3 | 58.6 | 19.0 | 45.7 | 20.4 | 54.0 | 19.2 | **48.9** | 6/20 |
| *Change InfoNCE loss to SK-InfoNCE loss* | | | | | | | | | | | | | | | | | | | | | | | |
| MUSE-SK | 14.4 | 39.9 | 16.5 | 44.2 | 19.7 | 49.5 | 26.4 | 58.6 | 13.2 | 34.0 | 19.1 | 52.5 | 19.5 | 49.4 | 26.8 | 59.3 | 17.6 | 46.6 | 20.1 | 54.3 | 19.3 | 48.8 | 7/20 |
| *Change STConv to STConv-GA* | | | | | | | | | | | | | | | | | | | | | | | |
| MUSE-SK-GA | 15.3 | 41.0 | 18.1 | 44.5 | 20.0 | 50.0 | 25.3 | 58.1 | 11.2 | 34.7 | 17.9 | 48.0 | 20.1 | 49.1 | 25.4 | 57.7 | 17.0 | 43.6 | 22.7 | 54.4 | 19.3 | 48.1 | 7/20 |

Table 3: Ablation Study of MUSE-Nerv series models, accuracy (%) of 200-way zero-shot classification: top-1 and top-5. The parts in bold represent the best results, while the underlined parts are the second best.

| Method | Subject 1 | | Subject 2 | | Subject 3 | | Subject 4 | | Subject 5 | | Subject 6 | | Subject 7 | | Subject 8 | | Subject 9 | | Subject 10 | | Ave | | Win |
| --- | --- | --- | --- | --- | --- | --- | --- | --- | --- | --- | --- | --- | --- | --- | --- | --- | --- | --- | --- | --- | --- | --- | --- |
| | top-1 | top-5 | top-1 | top-5 | top-1 | top-5 | top-1 | top-5 | top-1 | top-5 | top-1 | top-5 | top-1 | top-5 | top-1 | top-5 | top-1 | top-5 | top-1 | top-5 | top-1 | top-5 | subject score # |
| Subject dependent - train and test on one subject | | | | | | | | | | | | | | | | | | | | | | | |
| *Original MUSE-Nerv (NervFormer as EEG encoder & CLIP-ViT as image encoder with InfoNCE loss)* | | | | | | | | | | | | | | | | | | | | | | | |
| MUSE-Nerv | 11.0 | 33.9 | 12.3 | 37.4 | 13.6 | 39.4 | 19.1 | 48.0 | 10.7 | 31.9 | 14.0 | 41.2 | 13.0 | 41.3 | 21.0 | 54.6 | 15.4 | 38.6 | 17.1 | 43.9 | 14.7 | 41.0 | 0 |
| *Change InfoNCE loss to SK-InfoNCE loss* | | | | | | | | | | | | | | | | | | | | | | | |
| MUSE-SK-Nerv | 11.6 | 34.7 | 14.3 | 40.4 | 13.6 | 38.2 | 20.8 | 48.6 | 12.0 | 32.2 | 16.1 | 41.5 | 15.7 | 43.7 | 24.1 | 54.4 | 17.2 | 41.7 | 17.1 | 44.7 | 16.3 | 42.0 | 0 |
| *Change NervFormer to NervFormer-GA* | | | | | | | | | | | | | | | | | | | | | | | |
| MUSE-SK-Nerv-GA | 12.1 | 38.7 | 15.2 | 43.0 | 18.5 | 48.8 | 24.4 | 50.6 | 14.0 | 36.6 | 18.0 | 46.1 | 19.7 | 48.4 | 24.3 | 56.9 | 17.8 | 43.7 | 21.9 | 52.2 | 18.6 | 46.5 | 20/20 |

employed the Grad-CAM analysis method Selvaraju et al. (2016) to scrutinize our proposed best MUSE series models.

### 4.5.1 SPATIAL-TEMPORAL DYNAMICS ANALYSIS

To ensure that meaningful signals are preserved during Grad-CAM calculations, we take the absolute value of all Grad-CAM and EEG signal intensities of each trial for further analysis. The spatial-temporal comparison on both training and testing trials is depicted in Figure 7. We note that the higher-performing models, such as MUSE-SK and MUSE-SK-GA, concentrate on the EEG information between the 25th and 125th data points, corresponding to the 100 ms to 500 ms time period. Figure 8 illustrates a distinct response observed in the occipital cortex between 100 and 600 ms after the onset in MUSE-SK. However, the 200 ms stimulus onset asynchrony (SOA) continues to elicit periodic responses in the occipital cortex. Furthermore, a response in the parietal cortex is evident after 100 ms. This observation aligns with the bottom-up hierarchy of the visual system DiCarlo & Cox (2007), wherein visual stimuli are sequentially processed by V1, V2, and V4 in the occipital cortex, and subsequently by the inferotemporal region in the temporal cortex along the ventral stream for object recognition Bao et al. (2020).

## 5 CONCLUSION

In summary, this paper introduces the MUltimodal Similarity-keeping contrastivE learning (MUSE) framework, a novel approach tailored specifically for zero-shot EEG-based image classification, thereby addressing the intricate challenge of deciphering visual information from non-invasive EEG signals. Our method, drawing inspiration from established neuroscience findings, achieves state-of-the-art decoding accuracy, as substantiated by rigorous experimental evaluations. We further interpret our models and uncover insights into the spatial-temporal dynamics of EEG responses, shedding light on the neural processes underlying visual perception. We foresee that our work will catalyze further exploration in bridging the gap between EEG decoding and image recognition, advancing our understanding of visual cognition in the human brain.

## REFERENCES

Hamad Ahmed, Ronnie B Wilbur, Hari M Bharadwaj, and Jeffrey Mark Siskind. Object classification from randomized eeg trials. In *Proceedings of the IEEE/CVF Conference on Computer Vision and Pattern Recognition*, pp. 3845–3854, 2021.

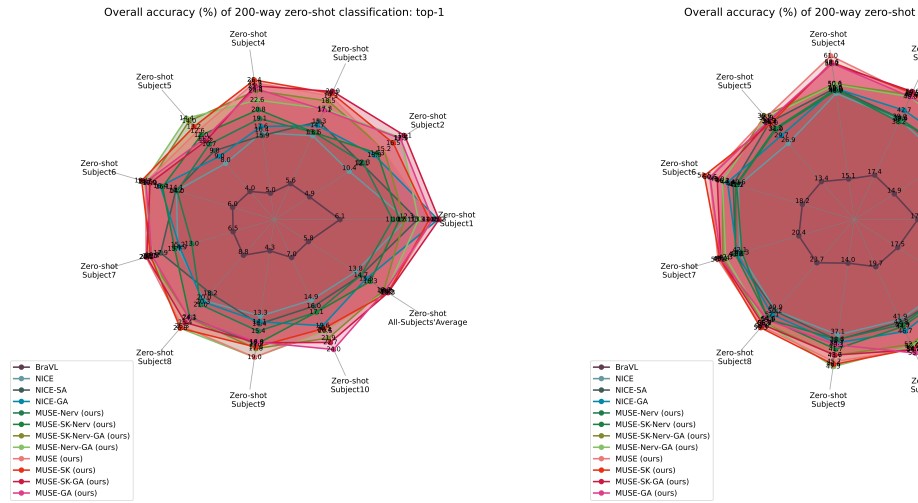

Figure 5: Overall Top-1 zero-shot accuracy comparison of all models.

Figure 6: Overall Top-5 zero-shot accuracy comparison of all models.

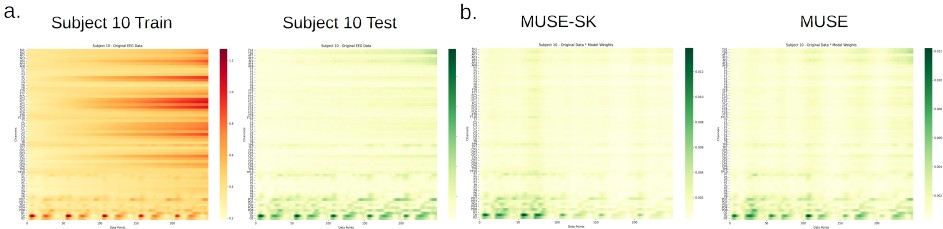

Figure 7: (a) Grad-CAM visualization of the MUSE series model averaged across all trials and repetitions for subject 10. (b) Comparative analysis reveals that MUSE-SK exhibits a heightened focus on the occipital lobes during the 100-500 ms time window compared to MUSE-SK and other models.

Pinglei Bao, Liang She, Mason McGill, and Doris Y Tsao. A map of object space in primate inferotemporal cortex. *Nature*, 583(7814):103–108, 2020.

Shaked Brody, Uri Alon, and Eran Yahav. How attentive are graph attention networks? In *The Tenth International Conference on Learning Representations (ICLR)*, 2022.

Sung-Yu Chen, Chi-Min Chang, Kuan-Jung Chiang, and Chun-Shu Wei. Ssvep-dan: A data alignment network for ssvep-based brain computer interfaces. *arXiv preprint arXiv:2311.12666*, 2023.

Ting Chen, Simon Kornblith, Mohammad Norouzi, and Geoffrey Hinton. A simple framework for contrastive learning of visual representations. In *International conference on machine learning*, pp. 1597–1607. PMLR, 2020.

James J DiCarlo and David D Cox. Untangling invariant object recognition. *Trends in cognitive sciences*, 11(8):333–341, 2007.

James J DiCarlo, Davide Zoccolan, and Nicole C Rust. How does the brain solve visual object recognition? *Neuron*, 73(3):415–434, 2012.

Changde Du, Kaicheng Fu, Jinpeng Li, and Huiguang He. Decoding visual neural representations by multimodal learning of brain-visual-linguistic features. *IEEE Transactions on Pattern Analysis and Machine Intelligence*, 2023.

Pascal Fries, John H Reynolds, Alan E Rorie, and Robert Desimone. Modulation of oscillatory neuronal synchronization by selective visual attention. *Science*, 291(5508):1560–1563, 2001.

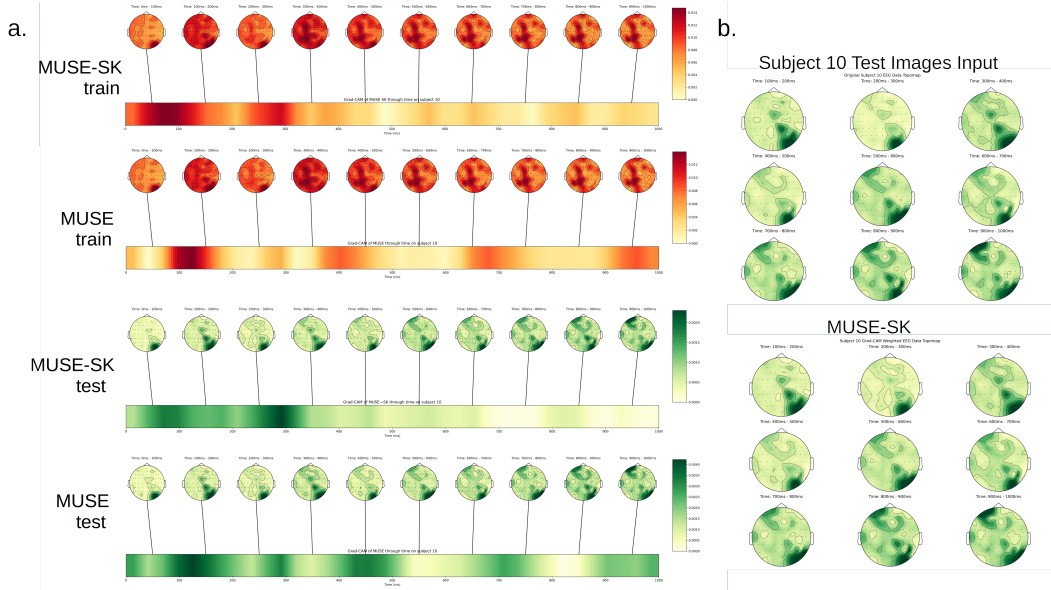

Figure 8: Topomap depicting the average response over each 100 ms interval across all trials, aggregated over all repetitions for subject 10. (a) Grad-CAM visualization for both MUSE-SK and MUSE models is presented, with the color bar at the bottom indicating the intensity of Grad-CAM for each model over time. Both models predominantly focus on the 100-500 ms time window. (b) A zoomed-in comparison between the input EEG data and the MUSE-SK model highlights the model's enhanced focus on temporal and occipital areas.

Kunihiko Fukushima. Neocognitron: A self-organizing neural network model for a mechanism of pattern recognition unaffected by shift in position. *Biological cybernetics*, 36(4):193–202, 1980.

Alessandro T Gifford, Kshitij Dwivedi, Gemma Roig, and Radoslaw M Cichy. A large and rich eeg dataset for modeling human visual object recognition. *NeuroImage*, 264:119754, 2022.

Jean-Bastien Grill, Florian Strub, Florent Altché, Corentin Tallec, Pierre Richemond, Elena Buchatskaya, Carl Doersch, Bernardo Avila Pires, Zhaohan Guo, Mohammad Gheshlaghi Azar, et al. Bootstrap your own latent-a new approach to self-supervised learning. *Advances in neural information processing systems*, 33:21271–21284, 2020.

Demis Hassabis, Dharshan Kumaran, Christopher Summerfield, and Matthew Botvinick. Neuroscience-inspired artificial intelligence. *Neuron*, 95(2):245–258, 2017.

Bin He, Lin Yang, Christopher Wilke, and Han Yuan. Electrophysiological imaging of brain activity and connectivity—challenges and opportunities. *IEEE transactions on biomedical engineering*, 58 (7):1918–1931, 2011.

Kaiming He, Haoqi Fan, Yuxin Wu, Saining Xie, and Ross Girshick. Momentum contrast for unsupervised visual representation learning. In *Proceedings of the IEEE/CVF conference on computer vision and pattern recognition*, pp. 9729–9738, 2020.

Martin N Hebart, Adam H Dickter, Alexis Kidder, Wan Y Kwok, Anna Corriveau, Caitlin Van Wicklin, and Chris I Baker. Things: A database of 1,854 object concepts and more than 26,000 naturalistic object images. *PloS one*, 14(10):e0223792, 2019.

Xin-Yao Huang, Sung-Yu Chen, and Chun-Shu Wei. Enhancing low-density eeg-based brain-computer interfacing with similarity-keeping knowledge distillation. *IEEE Transactions on Emerging Topics in Computational Intelligence*, 2023.

Alexander Ya Kaplan, Andrew A Fingelkurts, Alexander A Fingelkurts, Sergei V Borisov, and Boris S Darkhovsky. Nonstationary nature of the brain activity as revealed by eeg/meg: methodological, practical and conceptual challenges. *Signal processing*, 85(11):2190–2212, 2005.

Kendrick N Kay, Thomas Naselaris, Ryan J Prenger, and Jack L Gallant. Identifying natural images from human brain activity. *Nature*, 452(7185):352–355, 2008.

Wolfgang Klimesch. Eeg alpha and theta oscillations reflect cognitive and memory performance: a review and analysis. *Brain research reviews*, 29(2-3):169–195, 1999.

Yann LeCun, Léon Bottou, Yoshua Bengio, and Patrick Haffner. Gradient-based learning applied to document recognition. *Proceedings of the IEEE*, 86(11):2278–2324, 1998.

Yann LeCun, Yoshua Bengio, and Geoffrey Hinton. Deep learning. *nature*, 521(7553):436–444, 2015.

Dongjun Liu, Weichen Dai, Hangkui Zhang, Xuanyu Jin, Jianting Cao, and Wanzeng Kong. Brain-machine coupled learning method for facial emotion recognition. *IEEE Transactions on Pattern Analysis and Machine Intelligence*, 45(9):10703–10717, 2023. doi: 10.1109/TPAMI.2023.3257846.

Hesheng Liu, Yigal Agam, Joseph R Madsen, and Gabriel Kreiman. Timing, timing, timing: fast decoding of object information from intracranial field potentials in human visual cortex. *Neuron*, 62(2):281–290, 2009.

Weijian Mai, Jian Zhang, Pengfei Fang, and Zhijun Zhang. Brain-conditional multimodal synthesis: A survey and taxonomy. *arXiv preprint arXiv:2401.00430*, 2023.

Kyle E Mathewson, Alejandro Lleras, Diane M Beck, Monica Fabiani, Tony Ro, and Gabriele Gratton. Pulsed out of awareness: Eeg alpha oscillations represent a pulsed-inhibition of ongoing cortical processing. *Frontiers in psychology*, 2:99, 2011.

Yoichi Miyawaki, Hajime Uchida, Okito Yamashita, Masa-aki Sato, Yusuke Morito, Hiroki C Tanabe, Norihiro Sadato, and Yukiyasu Kamitani. Visual image reconstruction from human brain activity using a combination of multiscale local image decoders. *Neuron*, 60(5):915–929, 2008.

Aaron van den Oord, Yazhe Li, and Oriol Vinyals. Representation learning with contrastive predictive coding. *arXiv preprint arXiv:1807.03748*, 2018.

Yue-Ting Pan, Jing-Lun Chou, and Chun-Shu Wei. Matt: A manifold attention network for eeg decoding. *Advances in Neural Information Processing Systems*, 35:31116–31129, 2022.

Alec Radford, Jong Wook Kim, Chris Hallacy, Aditya Ramesh, Gabriel Goh, Sandhini Agarwal, Girish Sastry, Amanda Askell, Pamela Mishkin, Jack Clark, et al. Learning transferable visual models from natural language supervision. In *International conference on machine learning*, pp. 8748–8763. PMLR, 2021.

Maximilian Riesenhuber and Tomaso Poggio. Hierarchical models of object recognition in cortex. *Nature neuroscience*, 2(11):1019–1025, 1999.

Amanda K Robinson, Praveen Venkatesh, Matthew J Boring, Michael J Tarr, Pulkit Grover, and Marlene Behrmann. Very high density eeg elucidates spatiotemporal aspects of early visual processing. *Scientific reports*, 7(1):16248, 2017.

Guillaume A Rousselet, Jesse S Husk, Patrick J Bennett, and Allison B Sekuler. Single-trial eeg dynamics of object and face visual processing. *Neuroimage*, 36(3):843–862, 2007.

Jason Samaha and Bradley R Postle. The speed of alpha-band oscillations predicts the temporal resolution of visual perception. *Current Biology*, 25(22):2985–2990, 2015.

Paul S Scotti, Mihir Tripathy, Cesar Kadir Torrico Villanueva, Reese Kneeland, Tong Chen, Ashutosh Narang, Charan Santhirasegaran, Jonathan Xu, Thomas Naselaris, Kenneth A Norman, et al. Mindeye2: Shared-subject models enable fmri-to-image with 1 hour of data. *arXiv preprint arXiv:2403.11207*, 2024.

RR Selvaraju, M Cogswell, A Das, R Vedantam, D Parikh, and D Batra. Grad-cam: visual explanations from deep networks via gradient-based localization. arxiv [cs. cv], 2016.

Liang She, Marcus K Benna, Yuelin Shi, Stefano Fusi, and Doris Y Tsao. Temporal multiplexing of perception and memory codes in it cortex. *Nature*, pp. 1–8, 2024.

Prajwal Singh, Dwip Dalal, Gautam Vashishtha, Krishna Miyapuram, and Shanmuganathan Raman. Learning robust deep visual representations from eeg brain recordings. In *Proceedings of the IEEE/CVF Winter Conference on Applications of Computer Vision*, pp. 7553–7562, 2024.

Yonghao Song, Bingchuan Liu, Xiang Li, Nanlin Shi, Yijun Wang, and Xiaorong Gao. Decoding Natural Images from EEG for Object Recognition. In *The Twelfth International Conference on Learning Representations (ICLR)*, 2024.

Concetto Spampinato, Simone Palazzo, Isaak Kavasidis, Daniela Giordano, Nasim Souly, and Mubarak Shah. Deep learning human mind for automated visual classification. In *Proceedings of the IEEE conference on computer vision and pattern recognition*, pp. 6809–6817, 2017.

Yu Takagi and Shinji Nishimoto. High-resolution image reconstruction with latent diffusion models from human brain activity. In *Proceedings of the IEEE/CVF Conference on Computer Vision and Pattern Recognition*, pp. 14453–14463, 2023.

Yonglong Tian, Dilip Krishnan, and Phillip Isola. Contrastive multiview coding. In *Computer Vision–ECCV 2020: 16th European Conference, Glasgow, UK, August 23–28, 2020, Proceedings, Part XI 16*, pp. 776–794. Springer, 2020.

Praveen Tirupattur, Yogesh Singh Rawat, Concetto Spampinato, and Mubarak Shah. Thoughtviz: Visualizing human thoughts using generative adversarial network. In *Proceedings of the 26th ACM international conference on Multimedia*, pp. 950–958, 2018.

Shimon Ullman. Using neuroscience to develop artificial intelligence. *Science*, 363(6428):692–693, 2019.

Jose Antonio Urigüen and Begoña Garcia-Zapirain. Eeg artifact removal—state-of-the-art and guidelines. *Journal of neural engineering*, 12(3):031001, 2015.

Petar Veličković, Guillem Cucurull, Arantxa Casanova, Adriana Romero, Pietro Lio, and Yoshua Bengio. Graph attention networks. In *The Sixth International Conference on Learning Representations (ICLR)*, 2018.

Chun-Shu Wei and Tzyy-Ping Jung. Towards real-world neuromonitoring and applications in cognitive engineering. *Handbook of Neuroengineering*, pp. 3387–3404, 2023.

Chun-Shu Wei, Toshiaki Koike-Akino, and Ye Wang. Spatial component-wise convolutional network (sccnet) for motor-imagery eeg classification. In *2019 9th International IEEE/EMBS Conference on Neural Engineering (NER)*, pp. 328–331. IEEE, 2019.

Zesheng Ye, Lina Yao, Yu Zhang, and Sylvia Gustin. See what you see: Self-supervised cross-modal retrieval of visual stimuli from brain activity. *arXiv preprint arXiv:2208.03666*, 2022.

# A APPENDIX

## A.1 THE MODEL ABBREVIATIONS DETAILS

The abbreviations detail is shown as Table 4.

Table 4: The detail of all the model

| Method | EEG Encoder | Image Encoder | Loss Function |
|---|---|---|---|
| BraVL Du et al. (2023) | MLP | MLP | ELBO |
| NICE Song et al. (2024) | TSConv | CLIP-ViT | InfoNCE |
| NICE-SA Song et al. (2024) | TSConv-SA | CLIP-ViT | InfoNCE |
| NICE-GA Song et al. (2024) | TSConv-GA | CLIP-ViT | InfoNCE |
| MUSE (ours) | STConv | CLIP-ViT | InfoNCE |
| MUSE-GA (ours) | STConv-GA | CLIP-ViT | InfoNCE |
| MUSE-Nerv (ours) | NervFormer | CLIP-ViT | InfoNCE |
| MUSE-Nerv-GA (ours) | NervFormer-GA | CLIP-ViT | InfoNCE |
| MUSE-SK (ours) | STConv | CLIP-ViT | SK-InfoNCE |
| MUSE-SK-GA (ours) | STConv-GA | CLIP-ViT | SK-InfoNCE |
| MUSE-SK-Nerv (ours) | NervFormer | CLIP-ViT | SK-InfoNCE |
| MUSE-SK-Nerv-GA (ours) | NervFormer-GA | CLIP-ViT | SK-InfoNCE |

## A.2 GRAPH ATTENTION

In line with Graph Attention Networks (GATs) principles, we employ the Graph Attention (GA) module to iteratively refine the state of each node, conceptualized as electrodes, by leveraging the states of all other nodes Veličković et al. (2018); Brody et al. (2022). Through these mechanisms, the GA module dynamically adjusts the importance of each node based on the contextual information proffered by its neighbors, ensuring an attention-weighted update that underscores the interconnectivity of node features within the graph's architecture. Each node's representation is denoted by $n_i \in \mathbb{R}^{1 \times T}$, indexed by $i$ for $i = 1, \ldots, ch$, signifying an electrode that establishes connections with a defined set $\mathcal{N}_i$ of adjacent nodes, thus forming a fully connected graph. The update mechanism for an individual node $n_i$ is formalized as:

$$n_i' = \alpha_{i,i} W n_i + \sum_{j \in \mathcal{N}_i} \alpha_{i,j} W n_j \qquad (4)$$

where $n_i'$ designates the updated node, $\alpha_{i,j}$ encapsulates the attention coefficients indicative of the feature significance from node $j$ to node $i$, and $W$ is the weight matrix of the linear transformation. The attention coefficients $\alpha_{i,j}$ are computed via the equation:

$$\alpha_{i,j} = \frac{\exp(a^T \cdot \text{LeakyReLU}(W[n_i \| n_j]))}{\sum_{k \in \mathcal{N}_i \cup \{i\}} \exp(a^T \cdot \text{LeakyReLU}(W[n_i \| n_k]))} \qquad (5)$$

In this expression, $a \in \mathbb{R}^{2T}$ represents the weight vector of a feedforward attention mechanism, $()^T$ indicates the transpose operation, and $\|$ signifies concatenation. LeakyReLU is introduced as the non-linear function with a negative slope coefficient of 0.2, facilitating computational stability and non-linearity.

## A.3 TIME-FREQUENCY DYNAMICS ANALYSIS

We took the best SK model, MUSE-SK, to perform time-frequency analysis and found that the alpha wave, gamma wave, and theta wave signals were concentrated on the occipital and parietal lobes in both the training and testing topomaps. This finding aligns with medical literature, where the alpha wave is associated with visual attention Klimesch (1999); Mathewson et al. (2011), and the gamma wave is related to higher cognitive functions, attention, and visual processing Fries et al. (2001). This

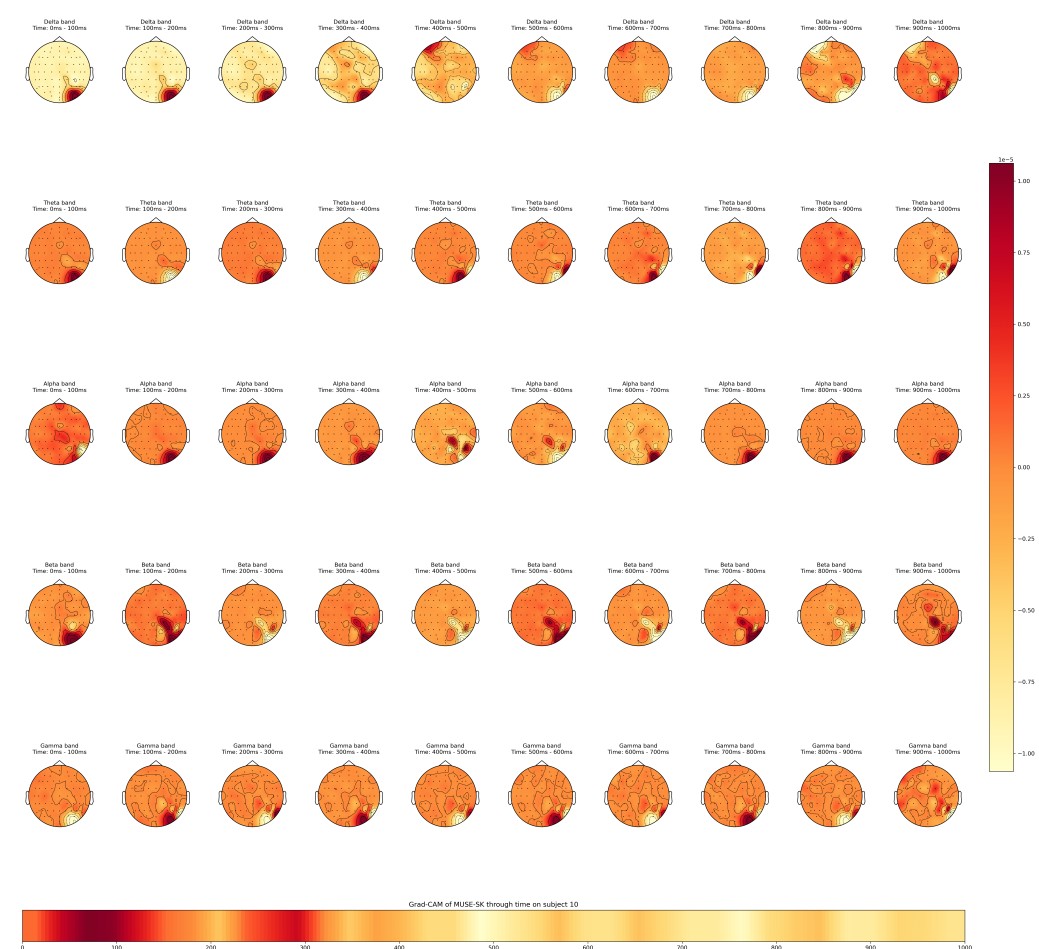

Figure 9: Time-Frequency map of MUSE-SK on averaging all of subject 10's training trials. We can see that the MUSE-SK can focus on alpha band and gamma band, where is related to vision attention and high-level visual recognition in neural science.

also indicates that our designed model has indeed learned some neural behaviors related to the human brain.

### A.4 LIMITATION

In our framework, we have not changed the image encoder to the more powerful CLIP, but we focus on comparing different EEG encoders under the same image encoder and the reliability of our proposed brain-inspired similarity-keeping framework. After demonstrating that this work can indeed improve the performance of contrastive learning, replacing the image encoder with a more powerful one would be a better direction.

### A.5 TABLE OF TESTING OBJECT CATEGORIES

We also try to use Grad-CAM method doing model interpretation on testing sets with our-selected category.

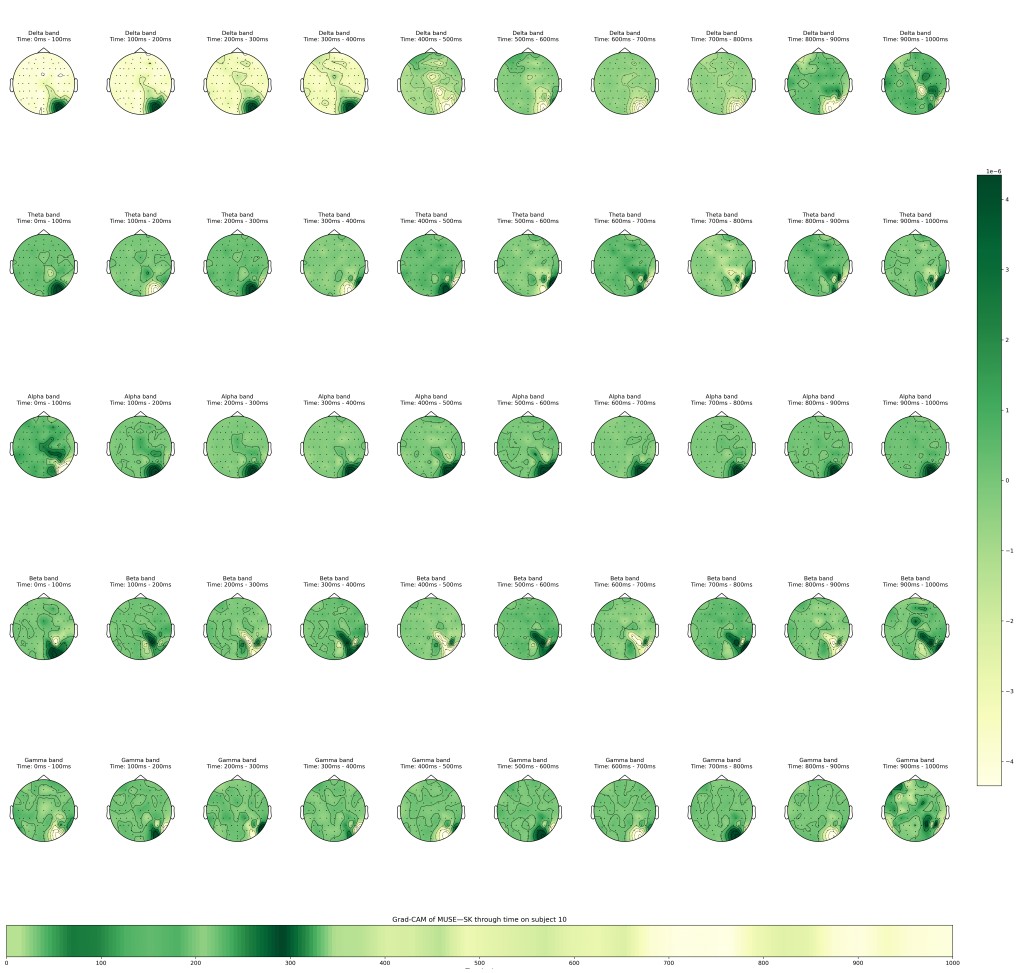

Figure 10: Time-Frequency map of MUSE-SK on averaging all trials in the testing set of subject 10. It is evident that MUSE-SK focuses on the alpha and gamma bands, which are associated with visual attention and high-level visual recognition in neuroscience.

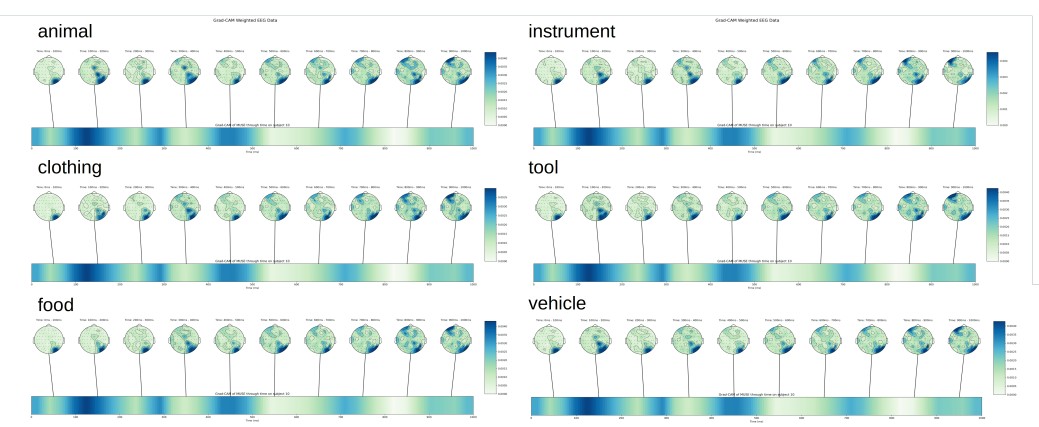

Figure 11: MUSE model interpretation on our-selected category.

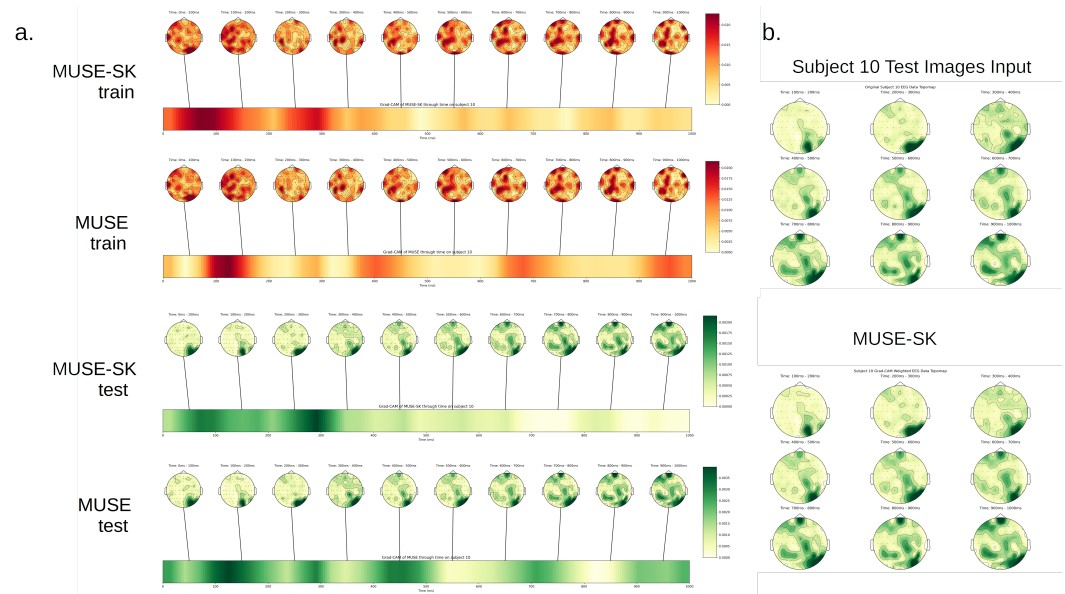

Figure 12: Topomap of each 100 ms by on one trial averaging through all the repetition on subject 10. (a.) On MUSE-SK and MUSE models, the color bar on the botton is the Grad-CAM of each model through time. Most of the model focus on the 100-500ms. The u (b.) Zoom-in and compare the input EEG data and the MUSE-SK, can see that the model can more focus on temporal and occipital areas.

Table 5: Test images on THINGSEEG dataset categories

| Category | Items |
|---|---|
| animal | 00002_antelope, 00012_beaver, 00024_bug, 00033_cat, 00034_caterpillar, 00039_cheetah, 00046_cobra, 00053_crab, 00058_crow, 00063_dalmatian, 00065_dragonfly, 00069_eagle, 00070_eel, 00072_elephant, 00076_flamingo, 00086_goose, 00087_gopher, 00088_gorilla, 00089_grasshopper, 00097_hummingbird, 00106_lamb, 00110_lightning_bug, 00111_manatee, 00117_mosquito, 00127_ostrich, 00129_panther, 00133_pheasant, 00136_pigeon, 00137_piglet, 00142_possum, 00144_pug, 00150_rhinoceros, 00152_rooster, 00161_seagull, 00183_tick, 00190_turkey |
| clothing | 00019_bonnet, 00037_chaps, 00043_cleat, 00045_coat, 00052_coverall, 00074_face_mask, 00083_glove, 00094_headscarf, 00096_hoodie, 00104_kneepad, 00107_lampshade, 00128_pajamas, 00138_pocket, 00155_sandal, 00169_snowshoe, 00176_suit, 00177_t-shirt, 00182_tiara, 00187_top_hat, 00189_tube_top |
| instruments | 00009_bassoon, 00041_chime, 00067_drum, 00080_french_horn, 00119_music_box, 00149_recorder |
| food | 00005_banana, 00007_basil, 00011_batter, 00015_birthday_cake, 00018_bok_choy, 00022_bread, 00027_bun, 00029_calamari, 00032_cashew, 00038_cheese, 00047_coconut, 00048_coffee_bean, 00050_cookie, 00051_cordon_bleu, 00054_creme_brulee, 00055_crepe, 00057_croissant, 00060_crumb, 00061_cupcake, 00064_dessert, 00071_egg, 00073_espresso, 00081_fruit, 00082_garlic, 00091_hamburger, 00098_ice_cube, 00101_jelly_bean, 00109_lettuce, 00112_marijuana, 00113_meatloaf, 00120_mussel, 00122_okra, 00123_omelet, 00124_onion, 00125_orange, 00126_orchid, 00131_pear, 00132_pepper1, 00135_pie, 00140_popcorn, 00141_popsicle, 00143_pretzel, 00147_radish, 00148_raspberry, 00157_sausage, 00158_scallion, 00159_scallop, 00162_seaweed, 00163_seed, 00174_strawberry, 00184_tomato_sauce, 00195_walnut, 00196_wheat, 00199_wine |
| tool | 00003_backscratcher, 00006_baseball_bat, 00016_blowtorch, 00020_bottle_opener, 00021_brace, 00023_breadbox, 00026_bullet, 00030_candlestick, 00035_cd_player, 00042_chopsticks, 00044_cleaver, 00049_coffeemaker, 00062_dagger, 00078_fork, 00079_freezer, 00090_grenade, 00092_hammer, 00093_handbrake, 00103_kettle, 00105_ladle, 00114_metal_detector, 00118_muff, 00130_paperweight, 00134_pickax, 00139_pocketknife, 00145_punch2, 00168_slingshot, 00170_spatula, 00171_spoon, 00173_stethoscope, 00185_tongs, 00186_tool, 00192_vise, 00197_wheelchair, 00200_wok |
| vehicle | 00001_aircraft_carrier, 00014_bike, 00017_boat, 00025_buggy, 00031_cart, 00059_cruise_ship, 00075_ferry, 00084_golf_cart, 00085_gondola, 00100_jeep, 00115_minivan, 00154_sailboat, 00160_scooter, 00164_skateboard, 00165_sled, 00172_station_wagon, 00175_submarine, 00191_unicycle |
| other | Other categories in test images. |

Figure 13: Time-Frequency map of MUSE-SK on one of subject 10's training trial. We can see that the MUSE-SK can focus on alpha band and gamma band, where is related to vision attention and high-level visual recognition in neural science.

Figure 14: Time-Frequency map of MUSE-SK on one trial in the testing set of subject 10. It is evident that MUSE-SK focuses on the alpha and gamma bands, which are associated with visual attention and high-level visual recognition in neuroscience.

