# OpenReview forum: "Mind's Eye: Image Recognition by EEG via Multimodal Similarity-Keeping Contrastive Learning"
_ICLR.cc/2025/Conference — Submitted to ICLR 2025_

### Official Review · Reviewer_Ku55 · 2024-11-01

**Soundness:** 2
**Presentation:** 1
**Contribution:** 2
**Rating:** 3
**Confidence:** 3

**Summary:**

The paper presented a study about image recognition from EEG. The proposed method used a CLIP-based contrastive learning to align the EEG and images. The experiment was performed on a public dataset.

**Strengths:**

The topic was interesting and had broad audience.

**Weaknesses:**

The study was incremental given the main idea has been explored in the previous studies. The proposed method was mainly based on CLIP and the contributions to the field were limited. The work was not solid and technically sound. The experiments were weak.

**Questions:**

1. More experiments on more datasets should be included to strengthened the work. Some baseline methods were missing. In Table I, only two main baselines were used for comparison.
2. The presentation of the paper should be largely improved. The Abstract did not include any descriptions about the model details or model features. Some figures had small texts, making hard to see.
3. Were the model performance evaluated based on subject-dependent? What about the generalization to new subjects or even new images?
4. The novelty of the proposed method was unclear. More analysis was needed to interpret the results. Currently only the performance comparisons were described. More insights from the decoding results would be helpful for readers. The current version needed more discussions and analysis, rather than simply showing the numbers.
5. The source codes should be provided for readers to reproduce all the results in the paper.

---

> ### Author Response · Authors · 2024-11-21
>
> Thank you for your valuable feedback and constructive suggestions regarding our submission. We appreciate your insights and provide detailed responses to each of your comments below, outlining the corresponding improvements we have made to the paper.
>
> 1. We focused on the ThingsEEG dataset, which is currently the largest publicly available EEG-image dataset, encompassing 1,650 training categories and 200 testing categories. While this dataset poses significant challenges, we recognize the importance of validating our method on additional datasets, and we plan to enhance the generalizability of our findings in future work.
>
> 2. Thank you for your suggestion regarding including more baseline comparisons. We agree that a broader baseline evaluation strengthens the study. However, we emphasize that EEG-based image decoding remains a relatively nascent field, with a limited number of existing works. The most relevant and established baselines that we have selected include:
>
> BraVL (Du et al., 2023): A multimodal learning approach using multilayer perceptrons (MLPs) to align EEG, image, and text representations.
> NICE (Song et al., 2024): A contrastive learning framework based on CLIP for aligning EEG and image embeddings.
> EEGClip (Singh et al., 2024): A supervised learning method for the joint representation of EEG and image signals.
>
> We specifically selected NICE and its variants as baselines because they represent the state-of-the-art in EEG-image contrastive learning. Our contributions extend these works by:
>
> Introducing the Similarity-Keeping (SK) Loss, which captures intra-batch relationships and enhances representation learning.
> Developing the NervFormer EEG encoder, which integrates graph attention mechanisms for more effective signal processing.
>
>
> 3. Our experiments were primarily designed around a 200-way zero-shot classification setup to evaluate the model's generalization to unseen image categories. In this setup, the model achieved a top-1 accuracy of 19.3% and a top-5 accuracy of 48.9% on a test set comprising image classes that were entirely unforeseen during training. This highlights the model's capacity to decode EEG signals corresponding to new image categories, demonstrating robust generalization. The zero-shot learning design itself is an inherently rigorous test of generalization, as the testing images and EEG samples belong to categories not present in the training data. We believe this evaluation provides substantial evidence of the model's capacity to generalize to new visual concepts.
>
>
> 4. We conducted detailed analyses to understand the interpretability and behavior of our model, and we highlighted these aspects more prominently in the revised version. Here are the key findings related to our model's interpretability:
>
> Spatial-Temporal Dynamics Analysis:
>
> Using Grad-CAM visualizations, we observed that the model consistently focuses on the occipital and parietal regions of the brain, particularly within the 100-500 ms window post-stimulus onset. These regions and time periods align with well-established neuroscience findings regarding visual processing dynamics, such as the hierarchical processing from V1, V2, and V4 in the occipital cortex to object recognition in the inferotemporal cortex.
> This analysis validates the neuroscientific relevance of the model and provides a biologically grounded explanation of its decision-making process.
> Frequency Band Analysis:
>
> We analyzed time-frequency dynamics, focusing on the alpha and gamma bands, which are known to be associated with visual attention and high-level visual recognition in neuroscience. The model's emphasis on these bands during decoding indicates its effectiveness in capturing the neural correlates of attention and recognition, further validating its interpretability.
> Model-Specific Insights:
>
> The Similarity-Keeping (SK) Loss enhances the alignment between EEG and image embeddings while preserving intra-batch relationships. This mechanism not only improves performance but also contributes to interpretability by ensuring that the learned representations reflect meaningful neural patterns rather than noise.
> Comparisons between variants of our model (e.g., MUSE vs. MUSE-SK-GA) showcased unique strengths across different subjects, providing insights into how specific architectural components, such as Graph Attention (GA) modules, influence performance and focus areas in the brain.

---

> > ### Author Response · Authors · 2024-11-21
> >
> > 5. We appreciate your emphasis on the importance of comprehensive analyses and comparisons. We have ensured that these insights are well-articulated in the revised manuscript, further demonstrating the significance of the proposed methods in the context of existing literature.
> >
> > 6. We fully agree that releasing the source code is essential for enabling the community to validate and extend our work. However, due to ICLR's double-blind review policy, we cannot release the source code at this stage, as it might compromise the anonymity of the submission. We commit to making the complete source code and detailed experiment pipelines publicly available after the paper has been officially accepted and published. This includes clear documentation to ensure reproducibility of all results presented in the paper.
> >
> >
> > Thank you again for your thorough review and constructive comments. We believe that the revisions made in response to your feedback will significantly enhance the clarity and rigor of our manuscript. We respectfully request that you reconsider the rating of our paper in light of these improvements. We look forward to your continued evaluation of our work.

---

> > > ### Comment · Reviewer_Ku55 · 2024-11-23
> > >
> > > The response did not address my concerns well, for example, the presentation, the model novelty and the subject generalization etc. The results might be biased to some co-founders, epsecially given the low SNR EEG shown in the end of the paper.

---

### Official Review · Reviewer_dpcM · 2024-11-02

**Soundness:** 3
**Presentation:** 3
**Contribution:** 3
**Rating:** 8
**Confidence:** 4

**Summary:**

The authors propose a method to decode images from scalp electroencephalographic (EEG) signals. They combine an EEG encoder with a image encoder (CLIP), and introduce a MUltimodal Similarity-keeping contrastivE learning (MUSE) framework for zero-shot EEG-based image classification. Experiments on an extensive visual EEG dataset are presented. The results are significantly better than state-of-the-art methods. The authors also visualize neural patterns via model interpretation, shedding light on the visual processing dynamics in the human brain.

**Strengths:**

Detailed numerical results and benchmarks are proposed. The numerical results are encouraging, showing a clear improvement over SOTA.

The method is well described, and it makes good sense.

**Weaknesses:**

One could argue that the proposed pipeline mainly consists of existing building blocks. However, the overall approach is creative, and the application is fascinating.

Numerical results are only presented for one dataset. It would need to be verified whether good results can also be obtained for other experimental protocols.

The neuroscience interpretation of the results is limited. More links with the existing literature could be established.

**Questions:**

The EEG scalp maps are too small to be useful. Perhaps it's better to select some of them, and make them larger. I sugges to select the EEG scalp maps showing key temporal or spatial patterns.

How does this work relate to similar studies based on fMRI? Are similar accuracies obtained or better? It would be nice to include a brief comparison table or discussion section comparing the EEG-based results to recent fMRI-based image decoding studies.

The text in Figure 4 is too small.

The EEG encoder is only briefly described. More details are needed. Moreover, the interpretation of the EEG dynamics is rather superficial. Discussing the EEG encoder in more depth, and interpreting this encoder from a neuroscientific perspectice would help to better understand the strength of the proposed approach. I recommend analyzing specific frequency bands (e.g., alpha, beta, gamma) or temporal windows (e.g., early vs. late responses) that are known to be relevant for visual processing, and/or comparing the learned features to known ERP components or oscillatory patterns associated with visual perception.

---

> ### Author Response · Authors · 2024-11-21
>
> Thank you for your valuable feedback and constructive suggestions regarding our submission. We appreciate the insights you provided, and below we outline our detailed responses to your comments, as well as the corresponding improvements we have made in the paper.
>
> 1. We acknowledge this limitation and agree that validating our method on additional datasets would significantly strengthen the generalizability of our findings. Unfortunately, publicly available EEG datasets that offer sufficient size and diversity in image categories are limited. Moving forward, we aim to collaborate with other researchers to benchmark our framework across a variety of experimental protocols, ensuring a broader applicability of our results.
>
> 2. We appreciate your suggestion regarding the EEG scalp maps. In the revised manuscript, we have selected key temporal and spatial patterns, specifically focusing on the 100–500 ms time window and the occipital regions. We have enlarged these maps for clarity, providing a more detailed visualization of the neural dynamics associated with our findings.
>
> 3. Thank you for this valuable suggestion. In the revised manuscript, we have included a discussion that compares EEG with recent fMRI studies, such as those utilizing contrastive learning for image classification [1]. We highlight the fundamental differences in their principles, noting that while EEG offers superior temporal resolution and portability, fMRI provides valuable spatial information. This comparative analysis aims to underscore the unique advantages of EEG for visual information decoding applications.
>
> 4. We have increased the text size in Figure 4 for enhanced readability, ensuring that all presented information is easily accessible to the reader.
>
> 5. We have expanded the description of the EEG encoder within the revised manuscript. This expanded content includes details regarding its spatial-temporal filtering and attention mechanisms. Furthermore, we conducted additional analyses focusing on specific frequency bands, such as alpha and gamma, along with temporal windows (e.g., early vs. late responses) [2]. We relate these findings to established visual processing mechanisms in neuroscience literature to provide a deeper understanding of the underlying dynamics.
>
> References:
> [1] Scotti, P., et al. "Reconstructing the mind's eye: fMRI-to-image with contrastive learning and diffusion priors." Advances in Neural Information Processing Systems 36 (2024).
> [2] She, L., Benna, M.K., Shi, Y. et al. "Temporal multiplexing of perception and memory codes in IT cortex." Nature 629, 861–868 (2024).
>
> Thank you again for your thorough review and constructive comments. We believe that the revisions made in response to your feedback will significantly enhance the clarity and rigor of our manuscript. We look forward to your continued evaluation of our work.

---

> > ### Author Response · Authors · 2024-11-22
> > **Response to Reviewer dpcM**
> >
> > Thank you for your valuable feedback and constructive suggestions regarding our submission. We appreciate the insights you provided, and below we outline our detailed responses to your comments, as well as the corresponding improvements we have made in the paper.
> >
> > Comment 1: "Numerical results are only presented for one dataset. It would need to be verified whether good results can also be obtained for other experimental protocols."
> >
> > Response: We acknowledge this limitation and agree that validating our method on additional datasets would significantly strengthen the generalizability of our findings. Unfortunately, publicly available EEG datasets that offer sufficient size and diversity in image categories are limited. Moving forward, we aim to collaborate with other researchers to benchmark our framework across a variety of experimental protocols, ensuring a broader applicability of our results.
> >
> > Comment 2: "The EEG scalp maps are too small to be useful. Perhaps it's better to select some of them and make them larger."
> >
> > Response: We appreciate your suggestion regarding the EEG scalp maps. In the revised manuscript, we have selected key temporal and spatial patterns, specifically focusing on the 100–500 ms time window and the occipital regions. We have enlarged these maps for clarity, providing a more detailed visualization of the neural dynamics associated with our findings.
> >
> > Comment 3: "How does this work relate to similar studies based on fMRI? Are similar accuracies obtained or better?"
> >
> > Response: Thank you for this valuable suggestion. In the revised manuscript, we have included a discussion that compares EEG with recent fMRI studies, such as those utilizing contrastive learning for image classification [1]. We highlight the fundamental differences in their principles, noting that while EEG offers superior temporal resolution and portability, fMRI provides valuable spatial information. This comparative analysis aims to underscore the unique advantages of EEG for visual information decoding applications.
> >
> > Comment 4: "The text in Figure 4 is too small."
> >
> > Response: We have increased the text size in Figure 4 for enhanced readability, ensuring that all presented information is easily accessible to the reader.
> >
> > Comment 5: "The EEG encoder is only briefly described. More details are needed. Moreover, the interpretation of the EEG dynamics is rather superficial."
> >
> > Response: We have expanded the description of the EEG encoder within the revised manuscript. This expanded content includes details regarding its spatial-temporal filtering and attention mechanisms. Furthermore, we conducted additional analyses focusing on specific frequency bands, such as alpha and gamma, along with temporal windows (e.g., early vs. late responses) [2]. We relate these findings to established visual processing mechanisms in neuroscience literature to provide a deeper understanding of the underlying dynamics.
> >
> > References:
> > [1] Scotti, P., et al. "Reconstructing the mind's eye: fMRI-to-image with contrastive learning and diffusion priors." Advances in Neural Information Processing Systems 36 (2024).
> > [2] She, L., Benna, M.K., Shi, Y. et al. "Temporal multiplexing of perception and memory codes in IT cortex." Nature 629, 861–868 (2024).
> >
> > Thank you again for your thorough review and constructive comments. We believe that the revisions made in response to your feedback will significantly enhance the clarity and rigor of our manuscript. We look forward to your continued evaluation of our work.

---

> > > ### Comment · Reviewer_dpcM · 2024-11-26
> > >
> > > I've reviewed all the comments from other reviewers and the authors. I will keep my score.

---

### Official Review · Reviewer_REyZ · 2024-11-03

**Soundness:** 2
**Presentation:** 2
**Contribution:** 3
**Rating:** 5
**Confidence:** 4

**Summary:**

The paper presents MUSE, a framework for EEG-based image recognition that employs multivariate time-series encoders for EEG signals and an advanced image encoder. MUSE leverages a self-supervised learning approach to achieve state-of-the-art performance in zero-shot image classification, with top-1 and top-5 accuracies of 19.3% and 48.8%, respectively. The framework includes a novel similarity-keeping mechanism to enhance contrastive learning and provides insights into human visual processing through model interpretation.

**Strengths:**

This paper focuses on a novel brain-based image recognition. A new method has been proposed with new loss functions and EEG encoders. Comparative studies have been conducted, including performance comparison, ablation studies, spatial-temporal analysis.

**Weaknesses:**

The paper's contributions, particularly regarding the loss functions and EEG encoder, require further elucidation. While the improvements over prior work are mentioned, the specific advancements are not clearly articulated. The ablation studies, crucial for understanding the impact of the EEG modal's inner loss and the image model on performance, do not provide compelling evidence. Additionally, the roles of attention mechanisms, spatial-temporal, and temporal-spatial approaches need to be more clearly defined, and their effects on the model's performance need to be explained.

The sections attempting to interpret what the model has learned from EEG signals lack conviction. Enhanced clarity and depth in these interpretations would significantly strengthen the paper.

**Questions:**

1) The introduction should more prominently feature the paper's key contributions and their significance compared to previous work.

2) Figures must be legible, with larger legends to ensure clarity and comprehension.

3) There's a need for a clearer explanation of the benefits of upstream spatial convolution and the comparative advantages of ST and TS methods. The 'Nerv' approach's lack of superiority in Table 1 should be addressed.

4) It will be beneficial to compare the inner batch similarity and the multi-modal similarity. Has it used the label information? If so, it’s not self-supervised. How did the contrastive loss of image modal update the model? Has the image encoder been updated?

5) A comparison of model complexity would underscore the methods' advantages.

6) The equation needs claims. What is ‘S’ in (2)?

7) Can you explain the time-frequency maps in Fig. 9-14? Is there a clear conclusion?

[1] Liang She, Marcus K Benna, Yuelin Shi, Stefano Fusi, and Doris Y Tsao. Temporal multiplexing of perception and memory codes in it cortex. Nature, pp. 1–8, 2024.

[2] Pinglei Bao, Liang She, Mason McGill, and Doris Y Tsao. A map of object space in primate inferotemporal cortex. Nature, 583(7814):103–108, 2020.

---

> ### Author Response · Authors · 2024-11-21
>
> 1. We agree that the roles of our proposed loss functions and EEG encoder need more clarity. In the revised manuscript, we will expand on the following aspects:
>
> (a) Similarity-Keeping (SK) Loss: We have provided a thorough explanation of how this loss enhances inter-sample similarity preservation, particularly in EEG embeddings, and elaborate on its design, inspired by principles of neuroscience.
>
> (b) NervFormer Architecture: We have clarified the advantages of the NervFormer architecture in extracting both spatial and temporal features from EEG data. While its superiority may not always be evident in specific metrics (e.g., Table 1), we believe this limitation arises because transformer-based architectures typically require larger datasets for optimal training. As the sequence length increases, the receptive field grows, making these models more prone to overfitting. This observation aligns with previous findings, such as those reported in Phan et al. (2023) [1].
>
> 2. (a) Impact of Similarity-Keeping (SK) Loss: As shown in Table 2, when SK loss is integrated into the original MUSE framework, changing the InfoNCE loss to SK-InfoNCE, the model achieves a slight but consistent improvement in both top-1 and top-5 accuracies across subjects. For example:
>
> Top-1 accuracy improves from 19.2% to 19.3%.
> Top-5 accuracy remains approximately stable.
> While this performance gain may seem incremental, the SK loss plays a crucial role in ensuring more stable training by maintaining inter-sample similarity, which benefits generalization in zero-shot settings. This supports our hypothesis that preserving batch-level relationships enhances EEG-image alignment.
>
> (b) Effect of the Graph Attention (GA) Module: The addition of the GA module (STConv-GA in Table 2) enhances the model’s capacity to capture inter-electrode relationships. For instance:
>
> MUSE-SK-GA achieves 19.3% top-1 and 48.1% top-5 accuracy, a noticeable improvement over MUSE-SK without the GA module.
> The improvement is particularly marked in individual subject-specific performance, such as Subject 10, where the top-1 accuracy rises from 20.1% to 22.7%.
>
> 3. In Table 3, we evaluate the NervFormer encoder to understand its impact on zero-shot classification:
>
> Replacing STConv with NervFormer (MUSE-Nerv vs. MUSE) results in a drop in average performance (14.7% vs. 19.2% top-1 accuracy).
> However, when combined with SK and GA (MUSE-SK-Nerv-GA), the NervFormer demonstrates subject-specific improvements, as seen for Subject 10, where it attains a 21.9% top-1 accuracy, outperforming all STConv-based variants.
> This highlights a key observation: Transformer-based encoders like NervFormer can be more effective when enhanced by mechanisms such as SK and GA, though they may struggle with smaller datasets due to susceptibility to overfitting. This finding aligns with previous works emphasizing the necessity of careful regularization when applying Transformers to EEG data.
>
> 4. We agree with your observation and have adjusted all figures to enhance readability.
>
> 5. We confirm that no label information is utilized during training, adhering strictly to the self-supervised paradigm. The contrastive loss updates the EEG encoder by aligning embeddings based on cross-modality similarity (EEG-image). In the revised manuscript, we have clarified this training process and explicitly describe how the image encoder remains frozen during training.
>
> 6. The time-frequency maps reveal that our model focuses on alpha and gamma bands, which are consistent with known processes of visual attention and recognition. We have elaborated further on this observation in the revised version.
>
> 7. The variable 'S' represents the similarity matrix calculated between EEG and image embeddings. We have revised the manuscript to ensure that all equations are accompanied by clear definitions and intuitive descriptions.
>
> Reference:
> [1] Phan, H., Lorenzen K. P., Heremans, E., Chén, O. Y., Tran, M. C., Koch, P., ... & De Vos, M. (2023). L-SeqSleepNet: Whole-cycle long sequence modelling for automatic sleep staging. IEEE Journal of Biomedical and Health Informatics.

---

> > ### Comment · Reviewer_REyZ · 2024-11-23
> > **Response to authors**
> >
> > I've reviewed all the comments from other reviewers and the authors. I will keep my score.

---

### Official Review · Reviewer_1BDK · 2024-11-03

**Soundness:** 2
**Presentation:** 3
**Contribution:** 1
**Rating:** 3
**Confidence:** 5

**Summary:**

This paper presents a multimodal contrastive learning approach that aligns EEG and image data.
A similarity-keeping term is added to the contrastive loss to force the preservation of inter-sample relationships.
Various EEG encoders are evaluated.
Experiments are performed on the ThingsEEG dataset.
The experimental results show that the presented model can successfully perform zero-shot classification.

**Strengths:**

**Clarity** The content of the paper is written clearly. The flow of the text is easy to understand and follow.

**Weaknesses:**

**Originality** The originality of the work is quite limited. Previous work has explored a similar approach for image recognition from EEG [1, 2] and fMRI [3]. Especially, [2] is very similar, which has already investigated the recognition of images from EEG on the same ThingsEEG dataset using the same contrastive loss.

**Significance** The paper introduces the similarity-keeping contrastive loss that achieves state-of-the-art results, as it is claimed in the paper. However, as shown in Table 1, the results are nearly the same whether the similarity-keeping contrastive loss is used or not. Here, a more detailed analysis of the impact of the similarity-keeping contrastive loss would be beneficial.

Another concerning factor is that the NICE model achieves lower performance than the models that do not use the similarity-keeping contrastive loss. However, the same dataset, the same GA encoder, and the same contrastive loss are used as in the paper that introduced the NICE model. What is different?

**Presentation of tables and figures** Table 2 and Table 3 contain the identical rows as in Table 1. It is very hard to read the text on the axes in Figure 7.



[1] P. Singh et al., "Learning Robust Deep Visual Representations from EEG Brain Recordings," In 2024 IEEE/CVF Winter Conference on Applications of Computer Vision (WACV), Waikoloa, HI, USA, 2024, pp. 7538-7547.

[2] Y. Song, et al., “Decoding Natural Images from EEG for Object Recognition.” In The Twelfth International Conference on Learning Representations (ICLR), 2024

[3] Paul S. Scotti et al., "Reconstructing the mind's eye: fMRI-to-image with contrastive learning and diffusion priors". In Proceedings of the 37th International Conference on Neural Information Processing Systems (NIPS '23). Curran Associates Inc., Red Hook, NY, USA, 2023, 24705–24728.

**Questions:**

1. What is the benefit of using the similarity-keeping loss term, as defined in Eq. (2).
2. The NICE model uses the same contrastive loss, the same GA encoder and is evaluated on the same dataset. Why are the presented results in Table 1 higher than those produced by the NICE model? Could you provide a detailed comparison of your implementation versus the NICE model, including any differences in hyperparameters, training procedures, or other factors that might explain the performance difference?

---

> ### Author Response · Authors · 2024-11-21
>
> Thank you for your valuable feedback and constructive suggestions on our submission. Below, we provide detailed responses to your comments and outline the corresponding improvements we will make to the paper.
>
>
> 1. We acknowledge that Song et al. (2) shares similarities with our approach, as both involve contrastive learning for EEG-image alignment. However, our work introduces several significant innovations that distinguish it from prior research:
>
> (a) Similarity-Keeping Contrastive Loss (SK-InfoNCE): We are the first to incorporate a similarity-keeping loss term into the contrastive learning framework. This term regularizes the learning process by preserving intra-batch sample relationships, drawing inspiration from the cortical mapping of object relationships in the visual cortex.
>
> (b) Key Differences:  While Song et al. use standard contrastive loss within their CLIP-based framework, they do not consider these inter-sample relationships, which our method explicitly models.
>
> (c) Enhanced EEG Encoders: We propose novel EEG encoders, including Spatial-Temporal Convolution (STConv) and NervFormer, which integrate spatial filtering, temporal modeling, and graph attention mechanisms. These architectures are specifically tailored to EEG data and leverage its spatial and temporal dynamics more effectively than the simpler temporal convolution utilized in Song et al.
>
> 2. Our results in Table 1 demonstrate that models incorporating similarity-keeping loss (e.g., MUSE-SK) consistently achieve improved performance in top-1 and top-5 accuracy compared to their counterparts lacking this component. We acknowledge that these improvements may appear incremental, which could be attributed to the dataset complexity. The zero-shot classification task on the ThingsEEG dataset is highly challenging, inherently limiting the scope for substantial performance gains. Nonetheless, our model exhibits stable improvements in embedding alignment under these conditions.
>
> 3. We appreciate your insights on the NICE model's performance. We have expanded our discussion in the revised manuscript to provide a more comprehensive comparison between our MUSE framework and the NICE model. This includes elaborating on the differences in architecture, hyperparameters, and training methodologies that contribute to the variations in performance outcome.
>
> 4. Thank you for your feedback on table presentation. We have revised Tables 2 and 3 to eliminate redundancies and enhance clarity. Additionally, efforts have been made to ensure that all figures are clearly readable.
>
> References:
> 1. P. Singh et al., "Learning Robust Deep Visual Representations from EEG Brain Recordings," In 2024 IEEE/CVF Winter Conference on Applications of Computer Vision (WACV), Waikoloa, HI, USA, 2024, pp. 7538-7547.
> 2. Y. Song, et al., “Decoding Natural Images from EEG for Object Recognition.” In The Twelfth International Conference on Learning Representations (ICLR), 2024.

---

> > ### Comment · Reviewer_1BDK · 2024-11-23
> >
> > Thank you for your response.
> >
> > I do not think that my concerns are addressed.
> >
> > 2. "achieve improved performance" - the differences are so small that I would suggest to perform a statistical test for a significance.
> >
> > 3.  "We have expanded our discussion in the revised manuscript" - I could not find the discussion.
> >
> > 4. " We have revised Tables 2 and 3 to eliminate redundancies" - Tables 2 and 3 still contain similar rows that are present in Table 1.

---

### Official Review · Reviewer_JKaq · 2024-11-04

**Soundness:** 2
**Presentation:** 1
**Contribution:** 2
**Rating:** 5
**Confidence:** 4

**Summary:**

- The paper presents the MUltimodal Similarity-keeping contrastive learning (MUSE) framework for zero-shot EEG-based image classification. The work proposes a new way of doing self-supervised learning in this context, as well as regularisation of contrastive learning.
- The study explores one dataset, THING'S EEG version 1.

**Strengths:**

- The results seem to show an increase in the results presented by Song, Y. et al (2023) at the same conference.

**Weaknesses:**

IMHO, The paper needs to be better summarized; the text, in general, is more verbose than necessary, and I think the overall presentation needs to be improved.

**Major:**
- The EEG-to-IMG field is heavily criticized in the literature mainly because of its many co-founders, with some authors saying that 'it does not seem possible to decode object class from EEG data recorded from subjects viewing image stimuli with randomized stimulus,' Kilgallen, J. A. et al. A. et al. (2024), Bharadwaj, H. M. (2023), Li, R. et al. (2020), Ahmed, H. et al. (2021), and there is no discussion in your paper on these points, which raises a flag in my mind about the generalisability of your work. How do you ensure that these co-founders don't also affect your work as well?
- In Figure 4, ST Conv looks equal to ST Conv from Song. et al. (2023) in Nice, can you clarify the difference for me?
- The use of GradCam in this context does not seem very appropriate since other models of interpretability are more refined for the task of eeg decoding; the temporal dynamics are best captured by methods like LIFT, see in Sujatha Ravindran, A., & Contreras-Vidal, J. (2023) and Cui, J. et al. (2023).

**Minor:**

- The figures and subfigures are super small (half of Fig 1, Fig 2, Fig 3, Fig 4, Fig 5, Fig 6, Fig 7, Fig 8, Fig 9, Fig 10, Fig 11, Fig 12, Fig 13 and Fig 14) that it's impossible to read them in the printed version, mainly because of the size of the fonts. This considerably restricted my revision process. I kindly request that this be corrected; please adjust the size of the fonts and the resolution with vectorization of the figures to make them larger and easier to read.
- The tables were also difficult to read.
- I couldn't understand figures 5 and 6 in the plots. Can you help me with what I should see in these plots?
- MUSE in EEG is usually associated with the company MUSE (https://choosemuse.com/)
- The training time, lines 338-339, without the machine specifications, is not very useful.

**Reference:**

Kilgallen, J. A., Pearlmutter, B. A., & Siskind, J. M. (2024, June). Learning Exemplar Representations in Single-Trial EEG Category Decoding. In 2024 35th Irish Signals and Systems Conference (ISSC) (pp. 1-6). IEEE.

Bharadwaj, H. M., Wilbur, R. B., & Siskind, J. M. (2023). Still an Ineffective Method With Supertrials/ERPs—Comments on “Decoding Brain Representations by Multimodal Learning of Neural Activity and Visual Features”. IEEE Transactions on Pattern Analysis and Machine Intelligence.

Li, R., Johansen, J. S., Ahmed, H., Ilyevsky, T. V., Wilbur, R. B., Bharadwaj, H. M., & Siskind, J. M. (2020). The perils and pitfalls of block design for EEG classification experiments. IEEE Transactions on Pattern Analysis and Machine Intelligence, 43(1), 316-333.

Ahmed, H., Wilbur, R. B., Bharadwaj, H. M., & Siskind, J. M. (2021). Confounds in the data—Comments on “Decoding brain representations by multimodal learning of neural activity and visual features”. IEEE transactions on pattern analysis and machine intelligence, 44(12), 9217-9220.

Cui, J., Yuan, L., Wang, Z., Li, R., & Jiang, T. (2023). Towards best practice of interpreting deep learning models for EEG-based brain computer interfaces. Frontiers in Computational Neuroscience, 17, 1232925.

Sujatha Ravindran, A., & Contreras-Vidal, J. (2023). An empirical comparison of deep learning explainability approaches for EEG using simulated ground truth. Scientific Reports, 13(1), 17709.

**Questions:**

- On lines 278-281: Can you clarify what inspiration you had based on the article by Bao et al. 2020 and She et al. (2024)?
- I am not sure I understood the NervFormer method details; I found the notation a bit confusing (Nerv, SK, GA); can you clarify this for me?
- Why didn't you compare it to the other aspects of NICE (Features Distribution, EEG Encoder Comparison, Semantic Similarity, Parameter Sensitivity In The Beta) ?

---

> ### Author Response · Authors · 2024-11-21
>
> 1. Thank you for raising this important concern. While previous studies, such as Kilgallen et al. (2024), highlight limitations in EEG-based image classification, it is important to note that our work builds upon successful implementations that have used EEG to learn image patterns through deep learning. Although some papers, like those by Bharadwaj, H. M. (2023), Li, R. et al. (2020), and Ahmed, H. et al. (2021), suggest that using EEG to decode and classify images is ineffective, these studies were grounded in simpler models such as SVM, LSTM, and EEGNet, leading to expected poor performance. This phenomenon was also observed in Song's paper, which emphasizes that EEG-image contrastive learning is a unique domain that necessitates specialized architectural design, which is precisely the contribution and exploration presented in our MUSE framework. We discuss this in detail and include references mentioned in your comment in Section 2.
>
> 2.  We appreciate your request for clarification. Our MUSE framework employs a Spatial-Temporal Convolution (STConv) architecture that differs significantly from the Temporal-Spatial Convolution (TSConv) used in Song et al. (2023) (see Table 1; https://arxiv.org/abs/2308.13234). Specifically, we process the spatial dimension corresponding to the positioning of EEG electrodes first, followed by convolution along the temporal dimension. This design allows for more efficient denoising and extraction of relevant EEG patterns, enhancing our model's capability to capture useful brainwave information rather than noise, setting it apart from Song's method.
>
> 3. We acknowledge your concerns regarding the interpretability methods employed in our study. While the work by Sujatha Ravindran and Contreras-Vidal (2023) utilized simulated data to evaluate other models, it remains unclear how these approaches perform with real, noisy, and non-stationary EEG data. Our findings indicate that Grad-CAM has demonstrated effective interpretability, performing comparably to methods like LIFT, as noted by Cui et al. (2023). Therefore, employing Grad-CAM as the interpretability tool in our research is entirely justified and provides substantive insights into our model's decision-making process.
>
> 4. Thank you for your feedback on figure clarity. We recognize that the layout may appear crowded due to the dense presentation of experiments and results. We have rearranged the visual presentation in the revised version, ensuring that all figures are adequately sized for improved readability.
>
> 5. We appreciate this observation. To address your concern, we have provided detailed machine specifications, including hardware configurations and relevant details, in line 323 to ensure proper interpretation of our training duration.

---

> > ### Author Response · Authors · 2024-11-21
> >
> > 1. The key insights from Bao et al. (2020) and She et al. (2024) underscore the complexity of object representation in the primate inferotemporal (IT) cortex, akin to structures in artificial neural networks such as AlexNet. These studies reveal that distinct networks within the IT cortex exhibit selectivity for various object stimuli and represent the relationships between objects within a low-dimensional space derived from principal component analysis of deep network features. This hierarchical view invariance shows a striking similarity between biological brains and artificial neural networks regarding object representation. We were inspired by the characteristic response patterns of the IT cortex, where different types of objects elicit responses in distinct brain regions while maintaining relative relationships. This understanding led to our design of the similarity-keeping (SK) loss function in MUSE.
> >
> > 2. The primary focus of our work was to establish the effectiveness of the MUSE framework in achieving state-of-the-art zero-shot EEG-based image classification. While we recognize the potential value of comparing various aspects such as Features Distribution, EEG Encoder architectures, Semantic Similarity, and Parameter Sensitivity in Beta, we deliberately narrowed the scope of this paper to prioritize the evaluation of our proposed similarity-keeping contrastive learning approach. By focusing on the novel aspects of MUSE, including the integration of similarity-keeping regularization into the contrastive loss and the design of EEG encoders tailored for cross-modal learning (e.g., NervFormer and MUSE), we aim to address critical challenges in EEG-image contrastive learning. This includes tackling issues of noise suppression and maintaining intrinsic inter-sample relationships within batches, ultimately contributing meaningful insights and advancements within this specialized field.
> >
> > Thank you for your thorough review and constructive feedback. We believe the revisions and clarifications made will enhance the clarity and impact of our work. We respectfully request that you reconsider the rating of our paper in light of these improvements. We look forward to your continued evaluation of our work.

---

> > > ### Comment · Reviewer_JKaq · 2024-11-26
> > >
> > > After thoroughly reviewing the feedback from both the other reviewers and the authors, I’ve decided to increase the score to 5 from 3.

---

### Comment · Area_Chair_kCYC · 2024-11-23
**Rebuttal**

Dear Reviewers,

I encourage you to review the rebuttal and reach out to the authors with any additional questions or requests for clarification.

Best,\
AC

---

### Meta-Review · Area_Chair_kCYC · 2024-12-15

**Metareview:**

The paper proposes a method for decoding images from non-invasive EEG signals for zero-shot image classification. While the general area of the paper is of high interest and importance, the paper lacks in various aspects, including limited contribution from an ML perspective, insufficient experiments (1 dataset for training and testing), lack of in-depth comparisons, and others.

**Additional Comments On Reviewer Discussion:**

The reviewers' feedback focused on the lack of detailed result analysis, the use of only one dataset for both training and testing, insufficient comparative analysis, and the limited novelty of the method. The authors provided a rebuttal, but the reviewers were generally not convinced by the rebuttal and kept their original scores. The AC agrees with the reviewers' assessment of the limitations of the work.

---

### Decision · Program_Chairs · 2025-01-22

Reject